# MathNAS: If Blocks Have a Role in Mathematical Architecture Design

**Qinsi Wang**[1*]     **Jinghan Ke**[1*]     **Zhi Liang**[2]     **Sihai Zhang**[3,4]

[1]University of Science and Technology of China
[2]School of Life Sciences, University of Science and Technology of China
[3]Key Laboratory of Wireless-Optical Communications, Chinese Academy of Sciences
[4]School of Microelectronics, University of Science and Technology of China
{wqs,jinghan}@mail.ustc.edu.cn, {liangzhi,shzhang}@ustc.edu.cn

## Abstract

Neural Architecture Search (NAS) has emerged as a favoured method for unearthing effective neural architectures. Recent development of large models has intensified the demand for faster search speeds and more accurate search results. However, designing large models by NAS is challenging due to the dramatical increase of search space and the associated huge performance evaluation cost. Consider a typical modular search space widely used in NAS, in which a neural architecture consists of $m$ block nodes and a block node has $n$ alternative blocks. Facing the space containing $n^m$ candidate networks, existing NAS methods attempt to find the best one by searching and evaluating candidate networks directly. Different from the general strategy that takes architecture search as a whole problem, we propose a novel divide-and-conquer strategy by making use of the modular nature of the search space. Here, we introduce MathNAS, a general NAS framework based on mathematical programming. In MathNAS, the performances of the $m * n$ possible building blocks in the search space are calculated first, and then the performance of a network is directly predicted based on the performances of its building blocks. Although estimating block performances involves network training, just as what happens for network performance evaluation in existing NAS methods, predicting network performance is completely training-free and thus extremely fast. In contrast to the $n^m$ candidate networks to evaluate in existing NAS methods, which require training and a formidable computational burden, there are only $m * n$ possible blocks to handle in MathNAS. Therefore, our approach effectively reduces the complexity of network performance evaluation. The superiority of MathNAS is validated on multiple large-scale CV and NLP benchmark datasets. Notably on ImageNet-1k, MathNAS achieves 82.5% top-1 accuracy, 1.2% and 0.96% higher than Swin-T and LeViT-256, respectively. In addition, when deployed on mobile devices, MathNAS achieves real-time search and dynamic network switching within 1s (0.4s on TX2 GPU), surpassing baseline dynamic networks in on-device performance. Our code is available at https://github.com/wangqinsi1/MathNAS.

## 1   Introduction

Neural Architecture Search (NAS) has notably excelled in designing efficient models for Computer Vision (CV) [1, 2, 3, 4] and Natural Language Processing (NLP) [5, 6, 7] tasks. With the growing popularity of the Transformer architecture [8, 9], designers are increasingly drawn to using NAS to

---

[*]Equal contribution.

37th Conference on Neural Information Processing Systems (NeurIPS 2023).

develop powerful large-scale models. Many existing NAS studies focus on designing search spaces for large models and conducting searches within them [10, 11, 12].

However, designing large models by NAS is challenging due to the dramatical increase of search space and the associated huge performance evaluation cost [10, 13]. Consider a widely used modular search space, in which a neural architecture is treated as a topological organization of $m$ different block nodes and each block node has $n$ different block implementations. Obviously, the number of possible networks or neural architectures, $n^m$, grows explosively with $n$ and $m$. In addition, candidate networks of large models are larger and require more computation for performance evaluation. Therefore, in order to conduct an effective architecture search, a proper search strategy and a suitable performance evaluation method are extremely important.

It is noteworthy that to improve search strategy, recent researches [14, 15] convert NAS to mathematical programming (MP) problems, which substantially decrease the search cost. MP-NAS provides a promising direction for rapidly designing large models. However, current MP-NAS methods exhibit certain architectural constraints. For example, DeepMAD [14] is solely applicable for architecture design within CNN search spaces, and LayerNAS [15] is exclusively suitable for hierarchically ordered search spaces. These limitations impede the application of MP-NAS methods to advanced search spaces, such as SuperTransformer [13, 5]. Besides, alternative strategies for effective performance evaluation of candidate networks are also expected, despite the improvement brought by parameter sharing [16, 17], performance prediction based on learning curves [18, 19] and so on.

In this study, we introduce MathNAS, a novel MP-NAS framework for universal network architecture search. In contrast to previous studies which estimate the performance of a network by solving a whole problem, MathNAS adopts an alternative divide-and-conquer approach. In brief, MathNAS improves the performance evaluation of a candidate network by estimating the performance of each block of the network first, and then combining them to predict the overall performance of the network. Although estimating block performance involves network training, predicting network performance is completely training-free. Therefore, this approach reduces the complexity of network performance evaluation. MathNAS achieves further improvement of the search strategy by transforming NAS to a programming problem, reducing the search complexity to polynomial time.

MathNAS contains three key steps:

- **Block performance estimation:** The performance of each block is estimated by the performance difference between networks having and having not that specific block.
- **Network performance prediction:** The performance of a network is predicted based on the performances of its blocks.
- **NAS by ILP:** Utilizing the block performances, NAS is solved as an Integer Linear Programming (ILP) problem.

We perform experiments on search spaces with various network architectures, including NAS-Bench-201 (GNN) [20], MobileNetV3 (CNN) [21], SuperTransformer (Transformer) [5] and NASViT (CNN+Trans) [13]. Our experiments demonstrate that predicting network performance based on its blocks' performances is applicable to different network architectures. In particular, the Spearman coefficient between the actual and the predicted top-1 indices on four different search spaces achieve 0.97, 0.92, 0.93, and 0.95, respectively. At the same time, by using the merit of the divide-and-conquer strategy to transform NAS into an ILP problem, MathNAS can find models superior to state-of-the-art (SOTA) models across different search spaces and tasks. In CV tasks, MathNAS achieves 82.5% top-1 accuracy on ImageNet-1k with 1.5G FLOPs and 15M parameters, outperforming AutoFormer-small (81.7%) and LeViT-256 (81.6%). In NLP tasks, MathNAS reaches a Blue Score of 28.8, on par with Transformer (28.4), but requiring only 1/5 of the FLOPs. In summary, our contributions are as follows:

1. We propose a general framework for performance evaluation of candidate networks by estimating block performance first and then combining them to predict network performance, which greatly improves the evaluation efficiency.

2. By virtue of the established mapping between block performance and network performance, we transform NAS into an ILP problem, which reduces the search complexity to polynomial.

3. We demonstrate MathNAS by considering three key performance indices for network design, i.e. accuracy, latency and energy, and achieve results superior to SOTA models.

## 2 The Proposed Method

**Search Space and Notations.** In this paper, we consider a widely used modular search space $\mathcal{S} = \{\mathcal{N}_1, ..., \mathcal{N}_k, ...\}$, in which network $\mathcal{N}_k$ consists of $m$ block nodes $\mathcal{B}_i (1 \leq i \leq m)$. For a block node, there is $n$ alternative blocks, i.e. $\mathcal{B}_i = \{b_{i,1}, ..., b_{i,j}, ..., b_{i,n}\}$, where block $b_{i,j} (1 \leq i \leq m, 1 \leq j \leq n)$ represents the $j$-th implementation of the $i$-th block node. A network can therefore be denoted as $\mathcal{N} = (b_{1,\mathcal{J}_1}, ..., b_{i,\mathcal{J}_i}, ..., b_{m,\mathcal{J}_m})$, with its $i$-th block node implemented by block $b_{i,\mathcal{J}_i}$. Totally, there are $m^n$ possible networks in the search space. Here, we focus on three key performance indices for network design, i.e. accuracy, latency and energy consumption. The accuracy, latency and energy consumption of network $\mathcal{N}$ are denoted as $Acc(\mathcal{N})$, $Lat(\mathcal{N})$, $Eng(\mathcal{N})$, respectively.

### 2.1 Problem Formulation: Reduce NAS Search Complexity from $\mathcal{O}(n^m)$ to $\mathcal{O}(m*n)$.

The objective of hardware-aware NAS is to find the best neural architecture $\mathcal{N}^*$ with the highest accuracy under limited latency $\hat{L}$ and energy consumption $\hat{E}$ in the search space $\mathcal{S}$:

$$\mathcal{N}^* = \arg\max_{\mathcal{N} \in \mathcal{S}} Acc(\mathcal{N}), s.t. Lat(\mathcal{N}) \leq \hat{L}, Eng(\mathcal{N}) \leq \hat{E} \tag{1}$$

In order to fulfill the above goal, NAS has to search a huge space, evaluate and compare the performance of candidate networks. Early NAS studies usually fully train the candidate networks to obtain their performance ranking, which is prohibitively time-consuming. Subsequent works introduce various acceleration methods. For instance, the candidate networks can avoid training from scratch by sharing weights [16, 17]. Also, the performance of a candidate network can be predicted based on the learning curve obtained from early termination of an incomplete training [18, 19]. Despite of these improvements, a candidate network has to be trained, no matter fully or partially, to obtain a reasonable performance evaluation. And the huge number of candidate networks in the search space poses a formidable efficiency challenge.

However, the modular nature of the search space may provide us with a novel possibility. Although there are $n^m$ candidate networks in the search space, they are all constructed by the $n*m$ blocks. If we can evaluate the performances of the blocks by training of a limited number of networks, and if we can combine these block performance indices to obtain a reasonable performance evaluation of networks, we can reduce the complexity of network performance evaluation from $\mathcal{O}(n^m)$ to $\mathcal{O}(n*m)$.

Guided by this idea, we reformulate the search of $\mathcal{N}^*$ from $\mathcal{S}$ as a succession of sub-problems. Each sub-problem corresponds to the task of searching the block $b_{i,\mathcal{J}_i^*}$ with the highest accuracy within the block node $\mathcal{B}_i$. This approach notably simplifies the original problem:

$$\mathcal{N}^* = (b_{1,\mathcal{J}_1^*}, b_{2,\mathcal{J}_2^*}, ..., b_{m,\mathcal{J}_m^*}) = \arg\max_{b_{1,j} \in \mathcal{B}_1}(b_{1,j}^A) \oplus \arg\max_{b_{2,j} \in \mathcal{B}_2}(b_{2,j}^A) \oplus \cdots \oplus \arg\max_{b_{m,j} \in \mathcal{B}_m}(b_{m,j}^A),$$
$$s.t. \sum_{i=1}^m b_{i,\mathcal{J}_i^*}^L \leq \hat{L}, \sum_{i=1}^m b_{i,\mathcal{J}_i^*}^E \leq \hat{E}, \tag{2}$$

where $b_{i,j}^A$, $b_{i,j}^L$, $b_{i,j}^E$ represent the accuracy, latency, and energy of block $b_{i,j}$ respectively. $\oplus$ denotes the operation of adding a block to the network. With this approach, each block $b_{i,j}$ is searched only once and the complexity can be effectively reduced to $\mathcal{O}(n*m)$. However, due to the mutual influence between blocks, a unified understanding of the relationship between the performance of $\mathcal{N}$ and its constituent blocks remains elusive, posing a challenge to the application of this method.

### 2.2 Divide-and-Conquer: Network-Block-Network

Consider the switching process $\mathcal{N}_{(i,1) \to (i,\mathcal{J}_i)}$, signifying the selection of the $i$-th $\mathcal{B}$ in network $\mathcal{N}$ as it switches from $b_{i,1}$ to $b_{i,\mathcal{J}_i}$, with other selected blocks remaining unchanged. Thus, any network $\mathcal{N} = (b_{1,\mathcal{J}_1}, b_{2,\mathcal{J}_2}, ..., b_{m,\mathcal{J}_m})$ can be viewed as the outcome of the base network $\widetilde{\mathcal{N}} = (b_{1,1}, b_{2,1}, ..., b_{m,1})$ undergoing $m$ sequential switching processes. Guided by this idea, we explore two aspects:

**E1: Can block performance be directly calculated as with network performance?**

Considering the entire search space $\mathcal{S}$, let us denote the collection of networks with the selected block $b_{i,1}$ as $\mathcal{N}_{(i,1)}^\Omega$, which comprises $n^{m-1}$ networks. For any network $\mathcal{N}_{(i,1)}$ in $\mathcal{N}_{(i,1)}^\Omega$, the switching process $\mathcal{N}_{(i,1) \to (i,j)}$ signifies the selection of the $i$-th $\mathcal{B}$ in network $\mathcal{N}$ as it switches from $b_{i,1}$

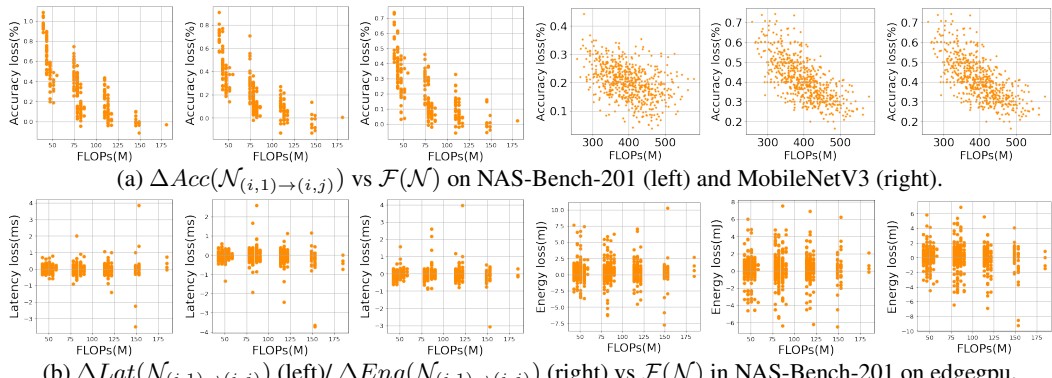

(a) $\Delta Acc(\mathcal{N}_{(i,1)\to(i,j)})$ vs $\mathcal{F}(\mathcal{N})$ on NAS-Bench-201 (left) and MobileNetV3 (right).

(b) $\Delta Lat(\mathcal{N}_{(i,1)\to(i,j)})$ (left)/ $\Delta Eng(\mathcal{N}_{(i,1)\to(i,j)})$ (right) vs $\mathcal{F}(\mathcal{N})$ in NAS-Bench-201 on edgegpu.

Figure 1: The relationship between $\Delta Acc(\mathcal{N}_{(i,1)\to(i,j)})$, $\Delta Lat(\mathcal{N}_{(i,1)\to(i,j)})$, $\Delta Eng(\mathcal{N}_{(i,1)\to(i,j)})$ and $\mathcal{F}(\mathcal{N})$. We conduct experiments on two search spaces: (1) NAS-Bench-201 [20]. Graph Network. We use the accuracy obtained on ImageNet after training each network independently. We count the accuracy of all networks in the search space. (2) MobileNetv3 [21]: sequentially connected CNN network. We use the accuracy obtained by the network with shared weights. We sample 5 blocks per block node and count the accuracies of 3125 subnetworks on the ImageNet validation set.

to $b_{i,j}$, with other selected blocks remaining unchanged. In this switch, two changes occur. The first change sees the selected block in $\mathcal{B}_i$ switching from $b_{i,1}$ to $b_{i,j}$. The second change arises from an internal adjustment in $\mathcal{B}_i$, modifying its interactions with other block spaces in the network. These changes lead to a difference in performance between $\mathcal{N}_{(i,1)}$ and $\mathcal{N}_{(i,j)}$, denote as $\Delta Acc(\mathcal{N}_{(i,1)\to(i,j)}), \Delta Lat(\mathcal{N}_{(i,1)\to(i,j)}), \Delta Eng(\mathcal{N}_{(i,1)\to(i,j)})$. By averaging performance difference obtained from all $n^{m-1}$ switching processes, $\mathcal{N}_{(i,1)\to(i,j)}^{\Omega}$, we can derive two key parameters:

1. $\Delta\phi(\mathcal{B}_{(i,1)\to(i,j)})$, the change in inherent capability of $\mathcal{B}_i$.
2. $\Delta\Phi(\mathcal{B}_{(i,1)\to(i,j)})$, the change in the interactive capability of $\mathcal{B}_i$ within $\mathcal{S}$.

Accordingly, we define the performances of $b_{i,j}$ as:

$$b_{i,j}^A = \overline{\Delta Acc\left(\mathcal{N}_{(i,1)\to(i,j)}^{\Omega}\right)} = \Delta\phi(\mathcal{B}_{(i,1)\to(i,j)}) + \Delta\Phi(\mathcal{B}_{(i,1)\to(i,j)}) \tag{3}$$

Similarly, $b_{i,j}^L$ and $b_{i,j}^E$ can be calculated employing the same methodology. The unbiased empirical validation and theoretical proof supporting this method can be found in the Appendix.

**E2: How can we predict network performance using block performance?**

To accurately derive the performance difference stemming from the switching process, we consider the performance difference that a particular block switch brings to different networks. We performed the identical process $\mathcal{N}_{(i,1)\to(i,\mathcal{J}_i)}$ over a range of networks within $\mathcal{N}_{(i,1)}^{\Omega}$. The outcome is illustrated in Figure 1, which depicts the relationship between the differences of three performances—latency, energy, and accuracy—and the FLOPs of the network. Our findings reveal that, within the same switching operation, latency and energy differences maintain consistency across different networks, while accuracy differences exhibit an inverse proportionality to the network's FLOPs, $\mathcal{F}(\mathcal{N})$.

This finding is logical, as networks with smaller FLOPs have lower computational complexity, rendering them more susceptible to block alterations. Conversely, networks with larger FLOPs exhibit higher computational complexity, making them less sensitive to individual block switching. In the Appendix, we proceed to fit the inverse relationship between accuracy difference and network FLOPs using various formulas. The results suggest that the accuracy difference is approximately inversely related to the network's FLOPs. Consequently, we can posit $\Delta Acc(\mathcal{N}_{(i,1)\to(i,\mathcal{J}_i)}) = \alpha * 1/\mathcal{F}(\mathcal{N})$, where $\alpha$ represents a specific block's coefficient.

Based on these observations, we can deduce that for any network $\mathcal{N}_{(i,1)}$ within the set $\mathcal{N}_{(i,1)}^{\Omega}$, the performance difference resulting from the switching process can be approximated as follows:

$$\Delta Lat(\mathcal{N}_{(i,1)\to(i,j)}) \approx b_{i,j}^L, \Delta Eng(\mathcal{N}_{(i,1)\to(i,j)}) \approx b_{i,j}^E, \Delta Acc(\mathcal{N}_{(i,1)\to(i,j)}) \approx b_{i,j}^A * \frac{\overline{\mathcal{F}\left(\mathcal{N}_{(i,j)}^{\Omega}\right)}}{\mathcal{F}(\mathcal{N}_{(i,j)})} \tag{4}$$

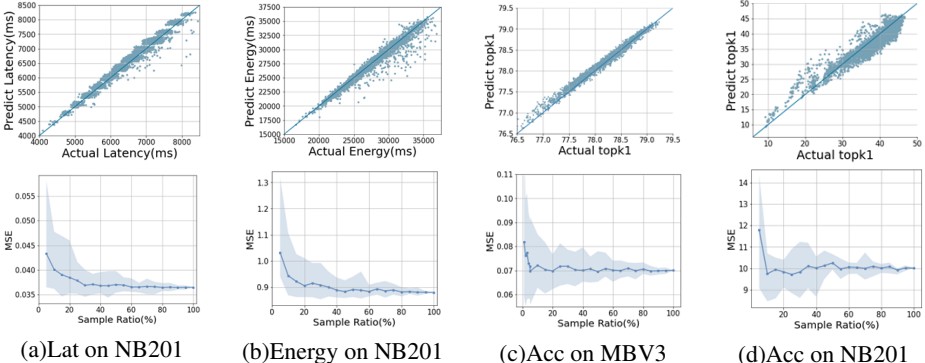

| (a)Lat on NB201 | (b)Energy on NB201 | (c)Acc on MBV3 | (d)Acc on NB201 |

Figure 2: Validation of the predictive performance of Equation 5 and the effect of sampling rate on it. The setup of the search space is the same as in Figure 1. The first row shows the predictive performance of Equation 5 when calculating $b_{i,j}^A$, $b_{i,j}^L$, and $b_{i,j}^E$ with Equation 3. The second row shows the predictive performance of Equation 5 when randomly sampling the network and computing the average difference to estimate $b_{i,j}^A$, $b_{i,j}^L$, and $b_{i,j}^E$ for the corresponding dataset.

By integrating Equation 3 and 4, we can estimate the performance of $\mathcal{N} = (b_{1,\mathcal{J}_1}, b_{2,\mathcal{J}_2}, ..., b_{m,\mathcal{J}_m})$:

$$Lat(\mathcal{N}) = Lat(\widetilde{\mathcal{N}}) - \sum_{i=1}^{m} b_{i,\mathcal{J}_i}^L \; , \; Eng(\mathcal{N}) = Eng(\widetilde{\mathcal{N}}) - \sum_{i=1}^{m} b_{i,\mathcal{J}_i}^E$$

$$Acc(\mathcal{N}) = Acc(\widetilde{\mathcal{N}}) - \sum_{i=1}^{m} b_{i,\mathcal{J}_i}^A * \frac{\overline{\mathcal{F}\left(\mathcal{N}_{(i,\mathcal{J}_i)}^\Omega\right)}}{\mathcal{F}(\mathcal{N}_{(i,\mathcal{J}_i)})} \tag{5}$$

**Proof-of-Concept Experiment.** In our examination of the predictive efficacy of Equation 5 across diverse network types, as depicted in Figure 2, we observe that it accurately forecasts three performances for both sequentially connected networks and graph networks and both weight-independent and weight-shared networks, all without necessitating network training. To the best of our knowledge, our method is the first work to precisely estimate network performances using a linear formula, and notably, it is theoretically applicable to all architectures.

### 2.3 Simplification: Single-sampling Strategy Instead of Full-sampling

Despite the outstanding predictive performance displayed by Equation 5, its computation of loss averages proves to be costly. In this section, by employing a single-sample sampling strategy, we effectively reduce the time complexity from $O(n^m)$ to $O(n*m)$, enhancing efficiency without compromising precision.

**Partial Network Sampling Strategies.** We begin by investigating the sample size requirements for Equation 5. The second row of Figure 2 demonstrates rapid convergence of Equation 5 with a notably small sample count for performance prediction. Specifically, Figure 2(c)(d) reveals that a mere 5% random sampling of networks is sufficient for the prediction to converge towards optimal performance. This underscores the impressive efficacy of Equation 5, which exhibits rapid convergence even with a limited number of random samples.

**Single Network Sampling Strategy.** Building upon Equation 4, we can select an average-FLOPs network, denoted as $\mathcal{N}^{avg}$, in the search

---

**Algorithm 1** Math Neural Architecture Search

**Stage1: Determine the Average-FLOPs Network**
**for** i=1,2,...,m **do**
 Calculate the average of the FLOPs of all blocks at the $i$-th note, $\overline{\mathcal{F}(b_i)} = (\mathcal{F}(b_{i,1}) + ... + \mathcal{F}(b_{i,n}))/n$;
 Select $b_{i,\mathcal{J}_i}$ whose FLOPs is closest to $\overline{\mathcal{F}(b_i)}$.
**end**
Define average-FLOPs net $\mathcal{N}^{avg} = \{b_{1,\mathcal{J}_1}, .., b_{m,\mathcal{J}_m}\}$.
**Stage2: Calculate Block Performances**
**for** i=1,2,...,m **do**
 **for** j=1,2,...,n **do**
  Switch the $i$-th block in $\mathcal{N}^{avg}$ from $b_{i,1}$ to $b_{i,j}$;
  Calculate the performance difference of the network brought about by switching and use it as the performance of the block $b_{i,j}^A$, $b_{i,j}^L$, $b_{i,j}^E$.
 **end**
**end**
**Stage3: Prediction and Architecture Search**
Calculate three characteristics of the base net $\widetilde{\mathcal{N}} = \{b_{1,1}, ..., b_{m,1}\}$ as $Acc(\widetilde{\mathcal{N}})$, $Lat(\widetilde{\mathcal{N}})$, $Eng(\widetilde{\mathcal{N}})$;
For network $\mathcal{N} = \{b_{1,\mathcal{J}_1}, b_{2,\mathcal{J}_2}, ..., b_{m,\mathcal{J}_m}\}$, its accuracy, latency and energy can be estimated by Equa. 5. Set required accuracy/latency/energy limit, and solve the corresponding ILP problem to obtain the architecture.

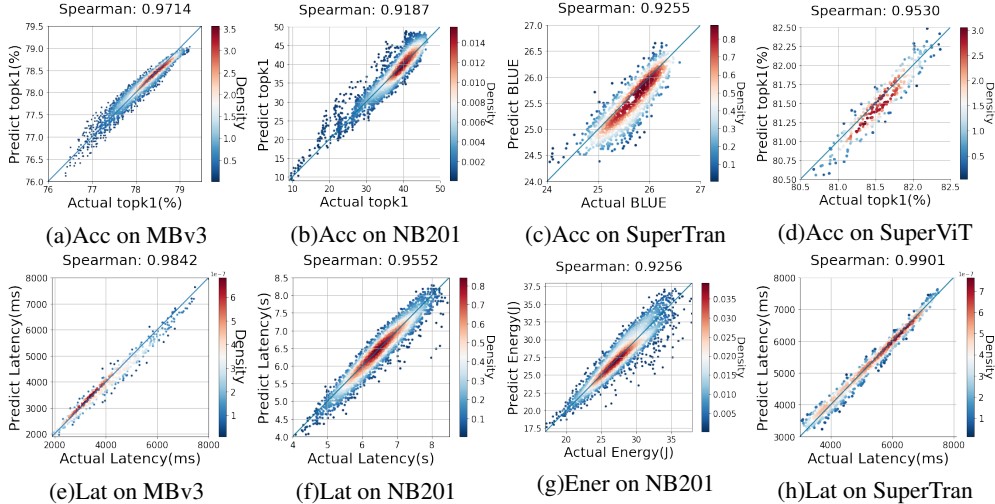

Figure 3: MathNAS algorithm verification. We conduct experiments on 4 search space. (1)Mo-bileNetV3 [21] (2)NAS-Bench-201 [20] (3)SuperTransformer [5] (4)SuperViT [13]. For NAS-Bench-201, we use the accuracy of each network trained individually as $Acc(\mathcal{N})$. For other spaces, the validate accuracy under shared weights base net is used. We show accuracy predictions on these four networks as well as hardware efficiency predictions on them. The calculation of $b_{i,j}^A, b_{i,j}^L$ and $b_{i,j}^E$ follows Algorithm 1. For NAS-Bench-201, we verify all nets and other spaces, we randomly sample 1000 nets to verify the prediction effect.

space with FLOPs approximating the mean value, ensuring $\overline{\mathcal{F}\left(\mathcal{N}_{(i,j)}^{\Omega}\right)}/\mathcal{F}(\mathcal{N}_{(i,j)}^{avg}) \approx 1$. This leads to: $b_{i,j}^A \approx \Delta Acc(\mathcal{N}_{(i,1)\to(i,j)}^{avg}), b_{i,j}^L \approx \Delta Lat(\mathcal{N}_{(i,1)\to(i,j)}^{avg}), b_{i,j}^E \approx \Delta Eng(\mathcal{N}_{(i,1)\to(i,j)}^{avg})$ By incorporating $\mathcal{N}^{avg}$ into Equation 5, we gain the ability to calculate any network performance. Thus, we only need to verify the performance of $\mathcal{N}_{(i,1)}^{avg}$ and $\mathcal{N}_{(i,j)}^{avg}$ for $b_{i,j}$, resulting in an $O(n*m)$ search time complexity. The complete MathNAS algorithm is presented in Algorithm 1, which, in theory, can achieve accurate prediction on any network structure with polynomial search complexity.

**Validation experiment of single-sample strategy effectiveness.** We assess the efficacy of Algorithm 1 across CNN, GCN, Transformer, and CNN+Transformer search spaces, with the outcomes displayed in Figure 3. It is evident that the Spearman correlation coefficient between predicted and actual values exceeds 0.9 for all networks. Remarkably, accuracies of 0.95 and 0.97 are attained on ViT and CNN architectures, respectively, emphasizing the algorithm's robustness.

## 2.4 MathNAS: Converting Architecture Search to ILP.

We denote $b_{i,j}^F$ as the FLOPs of block $b_{i,j}$, and $b_{i,j}^B \in \{0,1\}$ as the indicator of whether block $b_{i,j}$ is used in a network. If block $b_{i,j}$ is selected as the implementation of block node $\mathcal{B}_i$ in a network, $b_{i,j}^B = 1$, otherwise 0. The problem that NAS needs to solve can be formulated as:

$$\max_{b^B} Acc(\widetilde{\mathcal{N}}) - \frac{\sum_{i=1}^m \sum_{j=1}^n b_{i,j}^A * b_{i,j}^B}{\sum_{i=1}^m \sum_{j=1}^n b_{i,j}^F * b_{i,j}^B} * \overline{\mathcal{F}(\mathcal{N})}$$

$$s.t. \ Lat(\widetilde{\mathcal{N}}) - \sum_{i=1}^m \sum_{j=1}^n b_{i,j}^L * b_{i,j}^B \le \hat{L}, Eng(\widetilde{\mathcal{N}}) - \sum_{i=1}^m \sum_{j=1}^n b_{i,j}^E * b_{i,j}^B \le \hat{E}, \quad (6)$$

$$\sum_{j=1}^n b_{i,j}^B = 1, b_{i,j}^B \in \{0,1\}, \forall 1 \le i \le m.$$

The objective is to obtain the maximum accuracy network under two constraints. First, the latency and energy cannot exceed the limit. Second, for any block node, only one block is used. As Equation 6 is a fractional objective function, it can be transformed into an ILP problem by variable substitution.

# 3 Performance Evaluation

Experiments were conducted at three levels to evaluate the efficacy of MathNAS. Firstly, we validated the effectiveness of MathNAS on CV tasks by conducting searches across three different search spaces. Then we employed MathNAS to design efficient architectures for NLP tasks, showcasing its remarkable generalization capabilities. Finally, we leveraged MathNAS to perform real-time searches on edge devices, considering hardware resources, and achieved exceptional on-device performance.

## 3.1 Experimental Setup

**Search space.** For CV tasks, we validate our method on three search spaces: (1) NAS-Bench-201 [20]: a search space encompasses 15,625 architectures in a DARTS-like configuration. (2) SuperViT [13]: a hybrid search space that combines ViT and CNN, containing approximately $4 \times 10^{10}$ network architectures. (3) MobileNetV3 [21]: a lightweight network search space comprising about $10^{10}$ network architectures. For NLP tasks, we validate our approach on the SuperTransformer search space [5], which includes $10^{15}$ networks within a lightweight Transformer framework.

**Search and training settings.** For NAS-Bench-201 and MobileNetV3, we adopt the training methodology employed in [16] and [22] to train the base net for 100 epochs. Subsequently, we conducted MathNAS search on the base net. As for SuperTransformer and SuperViT, we adhere to the training algorithm proposed by [13] to train the base net for 100 epochs before conducting MathNAS search. The settings of hyperparameters in the training are consistent with the original paper. We employ the Gurobipy solver to address the ILP problem. In the SuperViT and SuperTransformer search spaces, we impose a search time limit of 10 seconds to expedite the process. For the other search spaces, we do not enforce any time constraints.

The search cost of MathNAS consists of two stages: offline network pre-training that is conducted only once and online real-time search. During the offline network pre-training, MathNAS evaluates block performance once. During online searching, MathNAS is capable of multiple real-time searches based on the current hardware resource constraints. To negate the influence of variations in GPU models and versions on the pre-training time, and to facilitate comparisons by future researchers, we have adopted pre-trained networks provided by existing works. All mentions of search cost in the paper refer solely to the real-time search time on edge devices. We provide a detailed description of the search space, more experimental results, and visualizations of the searched architectures in the Appendix.

## 3.2 MathNAS for Designing Effective CV Networks

**MathNAS for NAS-Bench-201.** To assess the effectiveness of our method in striking a balance between accuracy and hardware efficiency, we compare networks searched by MathNAS under hardware efficiency constraints to those searched by BRP-NAS [30], which utilizes GNN predictors to estimate network performance. As illustrated in Figure 4, MathNAS consistently locates networks that approach Pareto optimality in the majority of cases, whereas the employment of GNN predictors leads to suboptimal model choices. An extensive comparison between the searched architectures and SOTA models is provided in the Appendix for further insight.

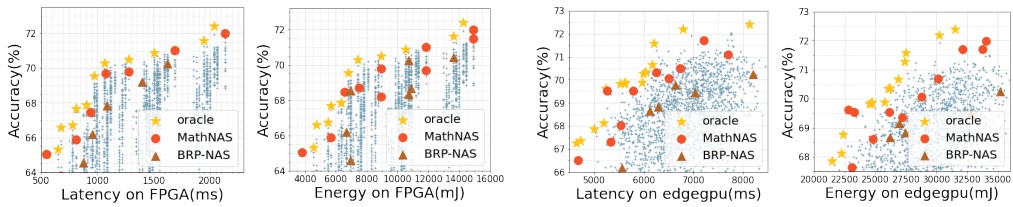

Figure 4: A comparison of networks searched by MathNAS (red circles) versus those searched by BRP-NAS (brown triangles) in the NAS-Bench-201 space. Across different devices, the networks searched by MathNAS demonstrate a closer proximity to the Pareto front (yellow five-pointed stars) as compared to the networks obtained through BRP-NAS.

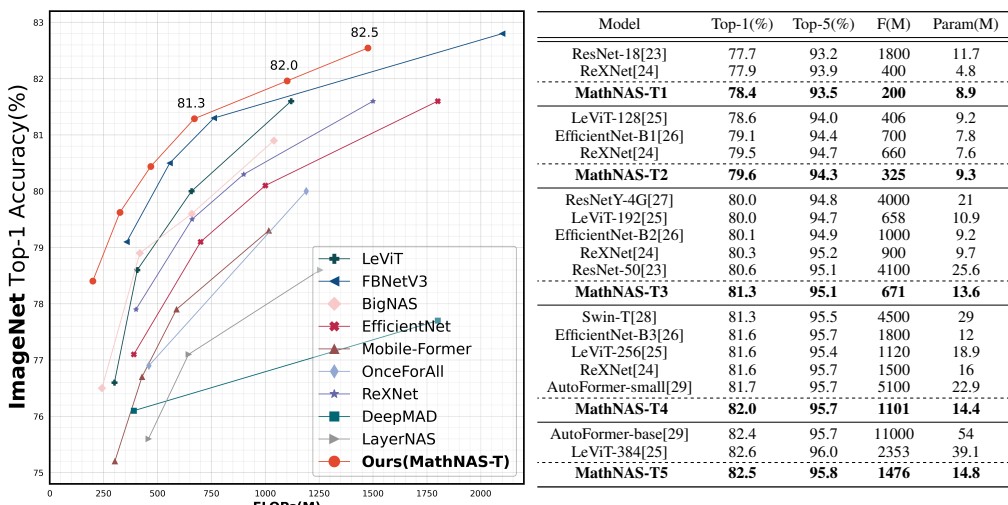

Figure 5: MathNAS v.s. SOTA ViT and CNN models on ImageNet-1K.

**MathNAS for ViT.** To assess the performance of MathNAS in designing larger models, we utilize it to create effective ViT models for ImageNet-1K classification. Figure 5 demonstrates that MathNAS surpasses or matches the performance of existing SOTA models. For instance, MathNAS-T5 achieves an accuracy of 82.5%, which is comparable to LeViT-384 [25] and Autoformer-base [29], while consuming only about 50% and 15% of their respective FLOPs. Similarly, MathNAS-T3 achieves comparable accuracy to RexNet [24] but with approximately half the FLOPs. MathNAS also exhibits exceptional performance in the realm of small networks. Particularly, MathNAS-T1 achieves a top-1 accuracy of 78.4%, surpassing ResNet-18 [23] and ReXNet [24] by 0.7% and 0.5% respectively.

**MathNAS for Mobile CNNs.** We employ MathNAS to design mobile CNN models for further investigation, conducting our search within the MobileNetV3 search space. As demonstrated in Table 1, MathNAS-MB4 achieves a top-1 accuracy of 79.2%, which is on par with EfficientNet-B1 (79.1%). It is important to note that EfficientNet-B1 is derived through a brute-force grid search, necessitating approximately 72,000 GPU hours [4]. Despite this, MathNAS-MB4 offers comparable performance to EfficientNet-B1 while only requiring 0.8 seconds to solve an ILP problem on the GPU and search for a suitable network. MathNAS also excels in the context of smaller networks. Notably, MathNAS-MB1 requires only 257M FLOPs to achieve a top-1 accuracy of 75.9%, surpassing the performance of FBNet-b [31], AtomNAS-A [32], and OFA [22], all of which demand higher computational resources.

Table 1: Performance of mobile networks designed with MathNAS. Top-1 accuracy on ImageNet-1K.

| Model | FLOPs(M) | Top-1 | Search Time | Scale Up |
|---|---|---|---|---|
| FBNet-b[31] | 295 | 74.1 | 609h | 1.9× |
| AtomNAS-A[32] | 258 | 74.6 | 492h | 2.3× |
| OFA[22] | 301 | 74.6 | 120h | 9.6× |
| **MathNAS-MB1** | **257** | **75.9** | **0.9s** | **4.6M×** |
| MnasNet-A1[33] | 312 | 75.2 | 40025h | 1.0× |
| ProxylessNAS-R[34] | 320 | 74.6 | 520h | 76.9× |
| AtomNAS-B[32] | 326 | 75.5 | 492h | 81.4× |
| FairNAS-C[35] | 321 | 71.1 | 384h | 104.2× |
| Single Path One-Shot[36] | 323 | 74.4 | 288h | 138.9× |
| OFA[22] | 349 | 75.8 | 120h | 333.5× |
| **MathNAS-MB2** | **289** | **76.4** | **1.2s** | **144M×** |
| EfficientNet B0[4] | 390 | 76.3 | 72000h | 1.0× |
| FBNet-c[31] | 375 | 74.9 | 580h | 124.1× |
| ProxylessNAS-GPU[34] | 465 | 75.1 | 516h | 139.5× |
| AtomNAS-C[32] | 363 | 76.3 | 492h | 146.3× |
| FairNAS-A[35] | 388 | 75.3 | 384h | 104.2× |
| FairNAS-B[35] | 345 | 75.1 | 384h | 104.2× |
| **MathNAS-MB3** | **336** | **78.2** | **1.5s** | **173M×** |
| EfficientNet B1[4] | 700 | 79.1 | 72000 | 1.0× |
| MnasNetA1[33] | 532 | 75.4 | 40025 | 1.8× |
| BigNAS-M[37] | 418 | 78.9 | 1152 | 62.5× |
| **MathNAS-MB4** | **669** | **79.2** | **0.8s** | **324M×** |

### 3.3 MathNAS for Designing Effective NLP Networks

We perform a comparative evaluation of MathNAS against SOTA NLP models on the WMT'14 En-De task to gauge its effectiveness. Table 2 reveals that MathNAS surpasses all baseline models in terms of BLEU score while also achieving FLOPs reduction across three distinct devices. Specifically, on Intel Xeon CPUs, MathNAS with full precision attains a remarkable 74% reduction in FLOPs compared to Transformer [8] and a 23% reduction compared to HAT [5], while registering improved BLEU scores by 0.4 and 0.3, respectively. Additionally, MathNAS excels in designing lightweight

Table 2: MathNAS vs. SOTA baselines in terms of accuracy and efficiency on NLP tasks.

| Model | Raspberry Pi | | | Intel Xeon CPU | | | Nvidia TITAN Xp GPU | | | Search Cost | |
| | FLOPs | BLEU | Latency | FLOPs | BLEU | Latency | FLOPs | BLEU | Latency | Time | CO$_2$ |
|---|---|---|---|---|---|---|---|---|---|---|---|
| Transformer[8] | 10.6G | 28.4 | 20.5s | 10.6G | 28.4 | 808ms | 10.6G | 28.4 | 334ms | 184h | 52lbs |
| Evolved Trans.[38] | 2.9G | 28.2 | 7.6s | 2.9G | 28.2 | 300ms | 2.9G | 28.2 | 124ms | 219200h | 624000lbs |
| HAT[5] | 1.5G | 25.8 | 3.5s | 1.9G | 25.8 | 138ms | 1.9G | 25.6 | 57ms | 200h | 57lbs |
| **MathNAS** | **1.7G** | **25.5** | **3.2s** | **1.8G** | **25.9** | **136ms** | **1.8G** | **25.9** | **68ms** | **10s** | **0.0008lbs** |
| HAT[5] | 2.3G | 27.8 | 5.0s | 2.5G | 27.9 | 279ms | 2.5G | 27.9 | 126ms | 200h | 57lbs |
| **MathNAS** | **2.1G** | **28.3** | **4.7s** | **2.4G** | **28.6** | **272ms** | **2.0G** | **28.1** | **107ms** | **10s** | **0.0008lbs** |
| HAT[5] | 3.0G | 28.4 | 7.0s | 3.5G | 28.5 | 384ms | 3.1G | 28.5 | 208ms | 200h | 57lbs |
| **MathNAS** | **2.8G** | **28.6** | **6.5s** | **2.8G** | **28.8** | **336ms** | **2.6G** | **28.6** | **189ms** | **10s** | **0.0008lbs** |

NLP models. On Nvidia TITAN Xp GPUs under latency constraints, MathNAS yields FLOPs comparable to HAT [5], but with a 0.3 higher BLEU score. A noteworthy aspect is that the network search process facilitated by MathNAS requires only 10 seconds, considerably reducing search time. As a result, employing MathNAS leads to a reduction of over 99% in $CO_2$ emissions compared to baseline models, underscoring its positive environmental impact.

## 3.4 MathNAS for Designing Dynamic Networks.

Deployed on edge devices (Raspberry Pi 4b, Jetson TX2 CPU, TX2 GPU), MathNAS allows dynamic network switching suited to device conditions. Within the MobileNetV3 search space, we account for memory limitations by calculating performance indices for each block, subsequently deploying five selected Pareto-optimal blocks balancing accuracy and latency in each block node. During runtime, latency is continuously monitored. Should it surpass a preset threshold, MathNAS immediately updates the blocks' latency. Then the device solves the ILP problem to identify the optimal network architecture, comparing and switching blocks with the searched network as required.

In comparison with SOTA dynamic network models, MathNAS demonstrates superior performance as outlined in Table 3 and Figure 6. Impressively, MathNAS solves the ILP problem on-device in a mere 0.4 seconds on the TX2 GPU, enabling real-time search. This notably enhances the number of executable networks on the device, outdoing SlimmableNet [39] and USlimmableNet [40] by factors of 781 and 116 respectively. Additionally, through a block-based approach, MathNAS enables efficient network alterations by replacing only essential blocks. When compared to Dynamic-OFA [41], which shares similar performance, MathNAS significantly reduces the switching time by 80%. The Appendix details the use of Pareto-optimal blocks and related network experiment results.

Table 3: MathNAS vs. SOTA baselines in terms of Dynamic Networks.

| Model | Network | | Latency (ms) | | | On Device Performance | | | | | |
| | Top-1 (%) | FLOPs (M) | Raspb Pi | TX2 CPU | TX2 GPU | Search Method | Search Time | Switch Unit | Switch Time | Nets Number | Scale Up |
|---|---|---|---|---|---|---|---|---|---|---|---|
| S-MbNet-v2[39] | 70.5 | 301 | 1346 | 958 | 118 | Manual Design | - | Channel | 15ms | 4 | 1.0x |
| US-MbNet-v2[40] | 71.5 | 300 | 1358 | 959 | 158 | Manual Design | - | Channel | 18ms | 27 | 6.7x |
| AS-MNASNet[42] | 75.4 | 532 | 2097 | 1457 | 2097 | Greedy Slimming | 4000h | Channel | 37ms | 4 | 1.0x |
| Dynamic-OFA[41] | 78.1 | 732 | 2404 | 1485 | 80 | Random+Evplution | 35h | Network | 244ms | 7 | 1.7x |
| **MathNAS** | **75.9** | **257** | **832** | **525** | **76** | **On-Device Search** | **0.4-12s** | **Block** | **61ms** | **3125** | **781x** |
| | **79.2** | **669** | **2253** | **1398** | **81** | | | | | | |

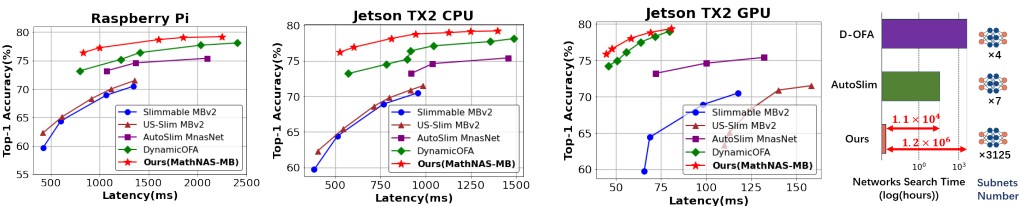

Figure 6: Top-1 vs. Latency of MathNAS over SOTA dynamic baselines on three devices.

# 4 Conclusion

This paper introduces MathNAS, the first architecturally-general MP-NAS. By virtue of the modular nature of the search space, we introduce block performance and establish the mapping from block performance and network performance, which enables subsequent transformation of NAS to an ILP problem. This novel strategy reduces network search complexity from exponential to polynomial levels while maintaining excellent network performance. We propose three transformation formulas to depict this process and support them with theoretical proofs and extensive experiments. MathNAS achieves SOTA performance on various large-scale CV and NLP benchmark datasets. When deployed on mobile devices, MathNAS enables real-time search and dynamic networks, surpassing baseline dynamic networks in on-device performance. Capable of conducting rapid searches on any architecture, MathNAS offers an appealing strategy for expediting the design process of large models, providing a clearer and more effective solution.

# 5 Acknowledgment

We thank the anonymous reviewers for their constructive comments. This work was partially supported by National Key R&D Program of China (2022YFB2902302). Sihai Zhang gratefully acknowledges the support of Innovative Research (CXCCTEC2200001).

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
