**Organization** In this supplementary file, we provide in-depth descriptions of the materials that are not covered in the main paper, and report additional experimental results. The document is organized as follows:

## A    Related Work

Neural Architecture Search (NAS) was introduced to ease the process of manually designing complex neural networks. Early NAS [1] efforts employed a brute force approach by training candidate architectures and using their accuracy as a proxy for discovering superior designs. Subsequent EA and RL-driven methods significantly enhanced search efficiency by sampling and training multiple candidate architectures [2, 3, 4]. One-shot NAS methods [5, 6, 7] further reduced the cost by training large supernetworks and identifying high-accuracy subnetworks, often generated from pre-trained models. Nevertheless, as search spaces expand with architectural innovations [8, 9], more efficient methods are necessary to predict neural network accuracy in vast design spaces.

Recent mathematical programming (MP) based NAS methods [10, 11] are noteworthy, as they transform multi-objective NAS problems into mathematical programming solutions. MP-NAS reduces search complexity from exponential to polynomial, presenting a promising avenue for large model architecture search. However, existing MP-NAS methods face architectural limitations. For instance, DeepMAD [10] is designed for convolutional neural networks, while LayerNAS [11] is suited for hierarchically connected networks. These limitations hinder MP-NAS usage in SOTA search spaces, leaving the challenge of swiftly designing effective large models unresolved. To address this, we propose an architecturally generalized MP-NAS, MathNAS. With the capability to perform rapid searches on any architecture, MathNAS presents an enticing approach to accelerate the design process for large models, providing a clearer and more effective solution.

## B    Theoretical Analysis

### B.1    $b_{i,j}^A$ Unbiased Proof

The main text introduces the definition of $b_{i,j}^A$ as follows:

$$b_{i,j}^A = \overline{\Delta Acc\left(\mathcal{N}_{(i,1)\rightarrow(i,j)}^{\Omega}\right)} = \frac{1}{n^{m-1}} \sum_{k=1}^{n^{m-1}} \left(Acc\left(\mathcal{N}_{(i,1)}^k\right) - Acc\left(\mathcal{N}_{(i,j)}^k\right)\right) \tag{1}$$

In investigating the unbiasedness of $b_{i,j}^A$, it is essential to first examine the distributions of $Acc\left(\mathcal{N}(i,1)\right)$ and $Acc\left(\mathcal{N}(i,j)\right)$. We conducted an experiment on NAS-Bench-201 to discern the distribution of accuracies for all networks that include a specific block. Figure 1 illustrates the experimental results, which suggest that, when the Kolmogorov-Smirnov (KS) test is applied to a beta distribution, all p-values exceed 0.05. This allows us to infer that the network accuracy related to a specific block adheres to a beta distribution, with the parameters $\alpha$ and $\beta$ varying among different blocks.

We can thus hypothesize that $Acc\left(\mathcal{N}(i,1)\right)$ and $Acc\left(\mathcal{N}(i,j)\right)$ follow the beta distributions with parameters $\alpha_1$, $\beta_1$ and $\alpha_2$, $\beta_2$, respectively. The expectation of $b_{i,j}^A$ is computed as follows:

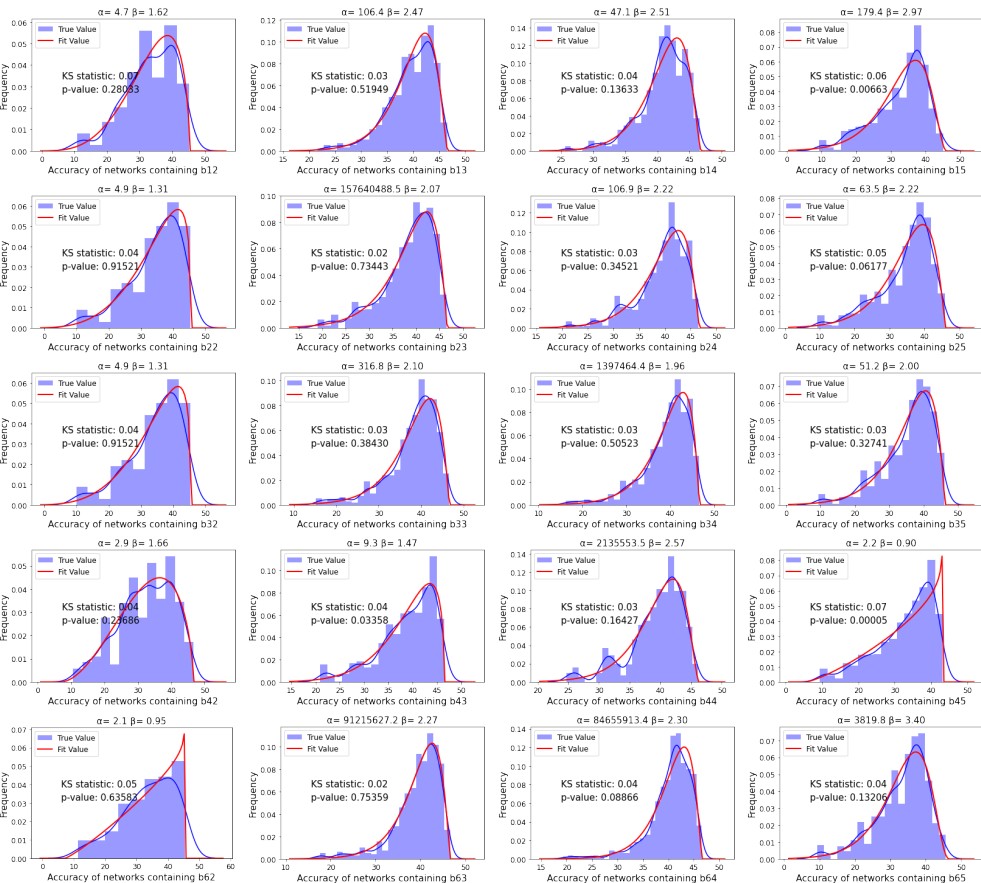

Figure 1: Distribution of accuracy for networks containing specific blocks on NAS-Bench-201. Accuracy is the result of the network training on ImageNet for 200 epochs. It can be seen that the ks verification $p$ value of the beta distribution on all blocks exceeds 0.05, so it can be considered that the accuracy of all networks containing a specific block is in line with the beta distribution.

$$E[b_{i,j}^A] = E[\frac{1}{n^{m-1}}\sum_{k=1}^{n^{m-1}}\left(Acc\left(\mathcal{N}_{(i,1)}^k\right) - Acc\left(\mathcal{N}_{(i,j)}^k\right)\right)]$$

$$= \frac{1}{n^{m-1}}\sum_{k=1}^{n^{m-1}}\left(E[Acc\left(\mathcal{N}_{(i,1)}^k\right)] - E[Acc\left(\mathcal{N}_{(i,j)}^k\right)]\right) \quad (2)$$

$$= E[Acc\left(\mathcal{N}_{(i,1)}\right)] - E[Acc\left(\mathcal{N}_{(i,j)}\right)]$$

$$= \frac{\alpha_1}{\alpha_1 + \beta_1} - \frac{\alpha_2}{\alpha_2 + \beta_2}$$

It is evident that the expected value of statistic $b_{i,j}^A$ represents the discrepancy between the expected values of $Acc\left(\mathcal{N}(i,1)\right)$ and $Acc\left(\mathcal{N}(i,j)\right)$, and is independent of any specific sample value. In other words, regardless of the sample values observed, the expected value of statistic $b_{i,j}^A$ always reflects the difference between the expected values of $Acc\left(\mathcal{N}(i,1)\right)$ and $Acc\left(\mathcal{N}(i,j)\right)$. Thus, we can assert that the statistic $b_{i,j}^A$ is unbiased, meaning the estimate of the difference between the expected values of $Acc\left(\mathcal{N}(i,1)\right)$ and $Acc\left(\mathcal{N}(i,j)\right)$ remains unskewed.

Similarly, it can be argued that $b_{i,j}^L$ and $b_{i,j}^E$ are also unbiased. This conclusion adds to our understanding of the general applicability of these statistics in analyzing network accuracy associated with specific blocks.

## B.2  Fit Validation

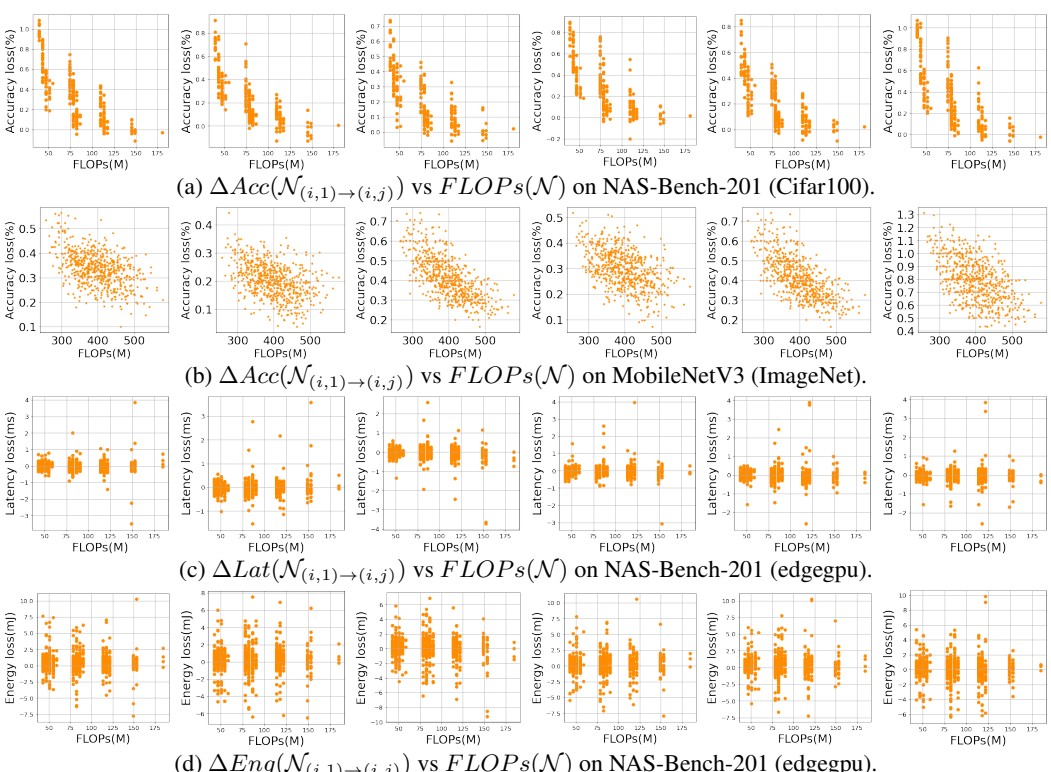

(a) $\Delta Acc(\mathcal{N}_{(i,1)\rightarrow(i,j)})$ vs $FLOPs(\mathcal{N})$ on NAS-Bench-201 (Cifar100).

(b) $\Delta Acc(\mathcal{N}_{(i,1)\rightarrow(i,j)})$ vs $FLOPs(\mathcal{N})$ on MobileNetV3 (ImageNet).

(c) $\Delta Lat(\mathcal{N}_{(i,1)\rightarrow(i,j)})$ vs $FLOPs(\mathcal{N})$ on NAS-Bench-201 (edgegpu).

(d) $\Delta Eng(\mathcal{N}_{(i,1)\rightarrow(i,j)})$ vs $FLOPs(\mathcal{N})$ on NAS-Bench-201 (edgegpu).

Figure 2: More result of the relationship between $\Delta Acc(\mathcal{N}_{(i,1)\rightarrow(i,j)})$, $\Delta Lat(\mathcal{N}_{(i,1)\rightarrow(i,j)})$, $\Delta Eng(\mathcal{N}_{(i,1)\rightarrow(i,j)})$ and $FLOPs(\mathcal{N})$ .

Within the main body of the text, we put forth several rules derived from our empirical observations:

1. For accuracy, $\Delta Acc(\mathcal{N}_{(i,1)\rightarrow(i,j_i)}) \approx \alpha * 1/\mathcal{F}(\mathcal{N})$.

2. For latency and energy consumption, $\Delta Lat(\mathcal{N}(i,1) \rightarrow (i,j_i))$ and $\Delta Eng(\mathcal{N}(i,1) \rightarrow (i,j_i))$ remain relatively constant across different networks when referring to a specific block.

We offer additional $\Delta Acc/Lat/Eng(\mathcal{N}(i,1) \rightarrow (i,j))$ - $FLOPs(\mathcal{N})$ pairs from NAS-Bench-201 and MobileNetV3 as depicted in Figure 2, which serve to substantiate the universality of these rules. Furthermore, we fit $\Delta Acc(\mathcal{N}(i,1) \rightarrow (i,j))$ - $FLOPs(\mathcal{N})$ using different inverse functions. As evidenced in Table 1, a reciprocal function provides the best fit for this relationship.

Table 1: Comparison of the fitting effect of different functions on the relationship between $\mathcal{N}_{(i,1)\rightarrow(i,j)}$ and $FLOPs(\mathcal{N})$.

| Net | Linear Function | | | Quadratic Function | | | Reciprocal Function | | | Log Function | | | Exp Function | | |
|---|---|---|---|---|---|---|---|---|---|---|---|---|---|---|---|
| | $R^2$ | MSE | DC | $R^2$ | MSE | DC | $R^2$ | MSE | DC | $R^2$ | MSE | DC | $R^2$ | MSE | DC |
| MobileNetV3 | 0.47 | 0.0061 | 0.47 | 0.45 | 0.0062 | 0.46 | 0.51 | 0.0059 | 0.48 | 0.48 | 0.0061 | 0.48 | 0.001 | 0.011 | 0.0011 |
| NAS-Bench-201 | 0.57 | 0.018 | 0.57 | 0.46 | 0.022 | 0.46 | 0.68 | 0.013 | 0.68 | 0.61 | 0.015 | 0.61 | 0.004 | 0.04 | 0.004 |

## B.3  Ablation Analysis

To elucidate the significance of each element within our predictive methodology, we instituted three sets of control models:

1. An accuracy prediction model that operates without FLOPs information.
2. A model predicting accuracy/latency where the Baseline network is arbitrarily chosen, instead of selecting a network with average FLOPs.
3. A model predicting accuracy/latency that uses any arbitrary block as the baseline block as opposed to the first block.

Table 2 illustrates the outcomes of these models. It becomes evident that the omission of any rule adversely affects the model's performance. The second model, in particular, proves to be the most vital. These findings corroborate the necessity for the basenet to be the network of average FLOPs, aligning with the rule between $\Delta Acc(\mathcal{N}_{(i,1)\rightarrow(i,j)})$ -$FLOPs(\mathcal{N})$ we derived earlier.

Table 2: Ablation Study of the proposed prediction method on MobileNetV3

| Predict Properties | Network | RMSE | MAE | Error |
|---|---|---|---|---|
| Accuracy | Complete Model | 0.117 | 0.0916 | - |
| | Without FLOPs info | 0.212 | 0.184 | 0.0924 |
| | Without Ave-FLOPs BaseNet | 0.513 | 0.416 | 0.324 |
| | Without Super Baseblock | 0.123 | 0.0992 | 0.0076 |
| Latency | Complete Model | 0.268 | 0.189 | - |
| | Without Ave-FLOPs BaseNet | 0.408 | 0.317 | 0.128 |
| | Without Super Baseblock | 0.276 | 0.194 | 0.005 |

## B.4  Effectiveness Analysis

This section provides an exploration into the efficacy of our proposed predictive model, substantiated by experimental validations. We draw comparisons with the classic GNN predictor on NAS-Bench-201 and MobileNetV3 search spaces. The GNN predictor's training process adheres to the Brp-NAS method. Figure 3 demonstrates that our model's MAE and MSE converge as the sample size of the network increases. This convergence is apparent in both weight-shared and weight-independent networks.

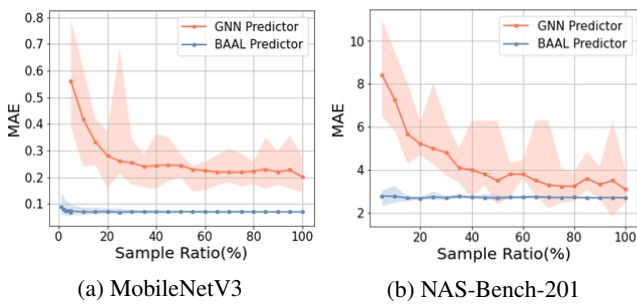

(a) MobileNetV3        (b) NAS-Bench-201

Figure 3: Sample effect of our accuracy predictor compared with GNN predictor.

## B.5 Space Generality Analysis

In our primary discussion, we reformulated the NAS problem as an Integer Linear Programming (ILP) problem. In this section, we delve further into the application of this formulation within the actual search space.

**Flexibility in Number of Block Nodes.** Some search spaces provide the flexibility to choose the number of block nodes. For instance, the SuperTransformer allows for the selection of anywhere between 1 and 6 decoders. For this kind of search space, MathNAS can also be applied. Specifically, for block node $i$ that has no block operation, we regard it as a special block of this node and represent it with $b_{i,0}$. Under this condition, the ILP formula can be modified accordingly as follows:

$$
\max_{b^B} Acc(\widetilde{\mathcal{N}}) - \frac{\sum_{i=1}^{m} \sum_{j=0}^{n} b_{i,j}^A * b_{i,j}^B}{\sum_{i=1}^{m} \sum_{j=1}^{n} b_{i,j}^F * b_{i,j}^B} * FLOPs(\overline{\mathcal{N}})
$$

$$
s.t.
$$

$$
Lat(\widetilde{\mathcal{N}}) - \sum_{i=1}^{m} \sum_{j=0}^{n} b_{i,j}^L * b_{i,j}^B \le \hat{L},
$$

$$
Eng(\widetilde{\mathcal{N}}) - \sum_{i=1}^{m} \sum_{j=0}^{n} b_{i,j}^E * b_{i,j}^B \le \hat{E}, \tag{3}
$$

$$
\sum_{k=1}^{i} b_{k,0}^B + \sum_{j=1}^{n} b_{i,j}^B = 1, b_{i,j}^B \in \{0,1\}, \forall 1 \le i \le m.
$$

**Applicability in Micro Search Space.** Unlike other search spaces we use, NAS-Bench-201 is a micro search space. In this paragraph, we explain how we performed our experiments in this search space and why MathNAS is applicable.

*Practical Implementation (How)*: During experiments within the NAS-Bench-201 space, we identify a set of edges in the same position across multiple GNN cells as a single "block". Given that the structure of GNN cells in the network remains consistent, our focus is on a single cell. Hence, any alteration in an edge operation essentially translates to a corresponding change in all cells, while other edge operations remain static.

*Theoretical Framework (Why)*: Regardless of the distinction between macro and micro search spaces, the networks in both are assembled from multiple mutable modules, be it blocks or edges. The capability of the entire network can be represented by the capabilities of these individual modules. In MathNAS, to explore the contribution of module capabilities to the network's performance, we evaluated changes in inherent module capabilities and their interactive capacities during module switches. This module evaluation methodology is applicable to the definition of blocks in NAS-Bench-201 that we proposed above: alterations in edge operations impact not only the specific edges' output data (inherent capability) but also influence the input and output data of other edges within the network (interactive capability). Therefore, MathNAS is theoretically suitable for micro search spaces represented by NAS-Bench-201.

## B.6 Searching Equation Solving Details

In this section, we describe in detail the solution of the fractional objective function programming equation proposed in the paper. The solution is divided into two steps.

1. Convert the original equation into an integer linear programming equation.
2. Solve the ILP equation.

**Equation Transformation.** In order to transform the equation into an ILP problem, we first perform variable substitution on the original equation.

$$
let \quad b_{i,j}^{\widetilde{B}} = \frac{b_{i,j}^B}{\sum_{i=1}^{m} \sum_{j=1}^{n} b_{i,j}^F * b_{i,j}^B}, \quad z = \frac{1}{\sum_{i=1}^{m} \sum_{j=1}^{n} b_{i,j}^F * b_{i,j}^B} \tag{4}
$$

Then the original equation can be transformed into the following integer linear programming problem:

$$O = \min_{b^B, z} \left( \sum_{i=1}^{m} \sum_{j=1}^{n} b_{i,j}^A * b_{i,j}^{\widetilde{B}} * \overline{\mathcal{F}(\mathcal{N})} \right)$$

$$s.t.$$

$$(Lat(\widetilde{\mathcal{N}}) - \hat{L}) * z \leq \sum_{i=1}^{m} \sum_{j=1}^{n} b_{i,j}^L * b_{i,j}^{\widetilde{B}}, (Eng(\widetilde{\mathcal{N}}) - \hat{E}) * z \leq \sum_{i=1}^{m} \sum_{j=1}^{n} b_{i,j}^E * b_{i,j}^{\widetilde{B}} \quad (5)$$

$$\forall 1 \leq i \leq m, \sum_{j=1}^{n} b_{i,j}^{\widetilde{B}} = z, b_{i,j}^{\widetilde{B}} \in \{0, z\}.$$

**ILP Solving.** To solve the ILP equations, we use the off-the-shelf Linprog Python package and the Gurobipy Python package to find feasible candidate solutions.

- Linprog is a basic integer programming solver that can be used on almost all edge devices, even on the resource-constrained Raspberry Pi. We use it to implement the branch and bound method and solve the ILP problem.

- Gurobipy is a more powerful solver, which has built-in a variety of advanced solving algorithms such as heuristic algorithms, and can flexibly utilize all available hardware resources on the device. Although Gurobipy is powerful, it requires more hardware resources than Linprog.

Therefore, for devices with limited hardware resources, we use Linprog for searching. For well-resourced devices, we use Gurobipy.

# C   Experimental Setups

## C.1   Description of Search Space

This section provides a detailed overview of the search spaces that we employ in our experimental setup. Furthermore, in our experiments, each resolution corresponds to a separate search space.

**NAS-Bench-201** search space encompasses cell-based neural architectures, where an architecture is represented as a graph. Each cell in this graph comprises four nodes and six edges, with each edge offering a choice among five operational candidates - zerorize, skip connection, 1-by-1 convolution, 3-by-3 convolution, and 3-by-3 average pooling. This design leads to a total of 15,626 unique architectures. The macro skeleton is constructed with one stem cell, three stages each composed of five repeated cells, residual blocks between the stages, and a final classification layer which incorporates an average pooling layer and a fully connected layer with a softmax function. The stem cell is formed by a 3-by-3 convolution with 16 output channels followed by a batch normalization layer. Each cell within the three stages has 16, 32 and 64 output channels respectively. The intermediate residual blocks contain convolution layers with stride 2 for down-sampling.

**MobileNetV3** search space follows a layer-by-layer paradigm, where the building blocks use MB-Convs, squeeze and excite mechanisms, and modified swish nonlinearity to construct efficient neural networks. This space is organized into five stages, each containing a number of building blocks varying from 2 to 4. The kernel size for each block can be chosen from {3, 5, 7} and the expansion ratio from {1, 4, 6}. The search space encapsulates approximately $10^{19}$ sub-nets, with each block offering 7,371 choices.

**SuperViT** search space is a tiled space featuring a fixed macroarchitecture. It is composed of three dynamic CNN blocks and four dynamic Transformer blocks connected in sequence. Each block allows for variations in width, depth, kernel size and expansion ratio. The SuperViT search space is displayed in Table 3.

Table 3: An illustration of SuperViT search space. Bold black represents blocks selected for the baseline net. The block in basenet corresponds to the largest of all available options.

| Block | Width | Depth | Kernel size | Expansion ratio | SE | Stride | Number of Windows |
|---|---|---|---|---|---|---|---|
| Conv | $\{\mathbf{16}, 24\}$ | - | 3 | - | - | 2 | - |
| MBConv-1 | $\{16, \mathbf{24}\}$ | $\{\mathbf{1}, 2\}$ | $\{\mathbf{3}, 5\}$ | 1 | N | 1 | - |
| MBConv-2 | $\{24, \mathbf{32}\}$ | $\{\mathbf{3}, 4, 5\}$ | $\{\mathbf{3}, 5\}$ | $\{4, 5, \mathbf{6}\}$ | N | 2 | - |
| MBConv-3 | $\{\mathbf{32}, 40\}$ | $\{3, 4, 5, \mathbf{6}\}$ | $\{\mathbf{3}, 5\}$ | $\{4, \mathbf{5}, 6\}$ | Y | 2 | - |
| Transformer-4 | $\{\mathbf{64}, 72\}$ | $\{3, 4, \mathbf{5}, 6\}$ | - | $\{1, 2\}$ | - | 2 | 1 |
| Transformer-5 | $\{\mathbf{112}, 120, 128\}$ | $\{3, 4, 5, \mathbf{6}, 7, 8\}$ | - | $\{1, 2\}$ | - | 2 | 1 |
| Transformer-6 | $\{160, 168, 176, \mathbf{184}\}$ | $\{3, 4, 5, \mathbf{6}, 7, 8\}$ | - | $\{1, 2\}$ | - | 1 | 1 |
| Transformer-7 | $\{\mathbf{208}, 216, 224\}$ | $\{3, 4, \mathbf{5}, 6\}$ | - | $\{1, \mathbf{2}\}$ | - | 2 | 1 |
| MBPool | $\{1792, \mathbf{1984}\}$ | - | 1 | 6 | - | - | - |
| Input Resolution | | | | $\{192, 224, 256, 288\}$ | | | |

**SuperTransformer** search space we employ is a large design space, constructed with Arbitrary Encoder-Decoder Attention and Heterogeneous Layers. This space contains an encoder layer and a decoder layer, the embedding dimensions of which can be selected from 512, 640. The encoder comprises six layers, each with an attention module and an FFN layer in series. The number of decoder layers can range from 1 to 6, and each decoder layer consists of two attention modules and one FFN layer in series. Each decoder has the option to focus on the last, second last or third last encoder layer. The hidden dimension of the FFN can be chosen from 1024, 2048, 3072, and the number of heads in attention modules from 4, 8. Every encoder layer has six choices, while each decoder layer provides 36 options. The search space and its corresponding baseline network are illustrated in Table 4.

Table 4: An illustration of SuperTransformer search space and the baseline net.

| Module | Choice | Baseline Net |
|---|---|---|
| encoder-embed | [640, 512] | 512 |
| decoder-embed | [640, 512] | 512 |
| encoder-ffn-embed-dim | [3072, 2048, 1024] | 2048 |
| decoder-ffn-embed-dim | [3072, 2048, 1024] | 2048 |
| encoder-layer-num | [6] | 6 |
| decoder-layer-num | [6, 5, 4, 3, 2, 1] | 6 |
| encoder-self-attention-heads | [8, 4] | 4 |
| decoder-self-attention-heads | [8, 4] | 4 |
| decoder-ende-attention-heads | [8, 4] | 4 |
| decoder-arbitrary-ende-attn | [-1, 1, 2] | 1 |

# D   Additional Experiments

## D.1   Experiment on NAS-Bench-201 Search Space

In the main body of this paper, we have showcased the competency of MathNAS in attaining accuracy-latency trade-offs within the NAS-Bench-201 search space. In this section, we extend our discussion to compare MathNAS with the current state-of-the-art models.

A comparative analysis with the leading-edge algorithms on NAS-Bench-201 is presented in Table 5. All the algorithms under consideration utilize the CIFAR-10 training and validation sets for architectural search and employ the NAS-bench-201 API to ascertain the ground-truth performance of the searched architecture across the three datasets. The reported experimental results are averaged over four separate searches.

From the results, it is evident that our method outperforms the state-of-the-art methods, with our best results approaching the peak performance. This robust performance serves as a testament to the efficacy of our proposed approach.

Table 5: Comparison results of MathNAS with state-of-the-art NAS methods on NAS-Bench-201.

| Method | CIFAR-10 | | CIFAR-100 | | ImageNet-16-120 | |
|---|---|---|---|---|---|---|
| | validation | test | validation | test | validation | test |
| DARTS[1st] | $39.77 \pm 0.00$ | $54.30 \pm 0.00$ | $15.03 \pm 0.00$ | $15.61 \pm 0.00$ | $16.43 \pm 0.00$ | $16.32 \pm 0.00$ |
| DARTS[2nd] | $39.77 \pm 0.00$ | $54.30 \pm 0.00$ | $38.57 \pm 0.00$ | $38.97 \pm 0.00$ | $18.87 \pm 0.00$ | $18.41 \pm 0.00$ |
| SETN | $84.04 \pm 0.28$ | $87.64 \pm 0.00$ | $58.86 \pm 0.06$ | $59.05 \pm 0.24$ | $33.06 \pm 0.02$ | $32.52 \pm 0.21$ |
| FairNAS | $90.07 \pm 0.57$ | $93.23 \pm 0.18$ | $70.94 \pm 0.84$ | $71.00 \pm 1.46$ | $41.9 \pm 1.00$ | $42.19 \pm 0.31$ |
| SGNAS | $90.18 \pm 0.31$ | $93.53 \pm 0.12$ | $70.28 \pm 1.20$ | $70.31 \pm 1.09$ | $44.65 \pm 2.32$ | $44.98 \pm 2.10$ |
| DARTS- | $91.03 \pm 0.44$ | $93.80 \pm 0.40$ | $71.36 \pm 1.51$ | $71.53 \pm 1.51$ | $44.87 \pm 1.46$ | $45.12 \pm 0.82$ |
| **Ours** | **90.18±0.00** | **93.31±0.00** | **71.74±0.00** | **70.82±0.00** | **46.00±0.00** | **46.53±0.00** |
| Optimal | 91.61 | 94.37 | 73.49 | 73.51 | 46.77 | 47.31 |

## D.2   Performance Comparison with Block-wise Methods

Existing blockwise methods such as DNA [12], DONNA [13], and LANA [14] use block distillation techniques to block the teacher model, obtaining the architecture of blocks to be replaced and their performance evaluation. Following this, they use the performance of each block to guide the algorithm in finding models with superior performance. Recent work [15] has pointed out the limitations of such methods. They depend on an excellent teacher network due to their use of distillation techniques. Furthermore, previous block performance estimation methods are unable to effectively predict actual block performance. Additionally, these methods are only suitable for the macro search space. In this section, we compare MathNAS with these methods and demonstrate that MathNAS overcomes these limitations.

We compare the performance of our method and previous blocking methods on the MobileNetV3 search space and the ImageNet dataset. For DNA and DONNA, we use the data reported in the original DONNA paper, where DONNA selects the data of the predictor after training on 40 samples.

In the case of LANA, due to the absence of the official code, we computed the metrics using the supernet in the search space as the teacher net, determining block evaluation scores from the variation in verification accuracy. The final network performance scores result from a linear addition, in line with LANA's methodology. The results are shown in Table 6. Evidently, MathNAS outperforms prior block-wise methods in both network performance evaluation and accuracy prediction.

Table 6: Comparison of prediction accuracy between our proposed method and block-wise methods on the MobileNetV3 (x 1.2) search space with the ImageNet dataset.

| Methods | Kendall-Tau | MSE | Block Evaluation | Net Evaluation |
|---------|-------------|-----|------------------|----------------|
| DNA | 0.74 | NA | Block Knowledge Distillation | Sorting Model |
| DONNA | 0.82 | 0.08 | Distillation Loss | Linear Regression |
| LANA | 0.77 | 0.04 | Change of Validation Accuracy | Simple Summation |
| **MathNAS** | **0.86** | **0.01** | **Average of Accuracy Variations** | **MathNAS Formula** |

Previous blocking methods were targeted at macrospaces, while MathNAS has excellent performance in both macrospaces and microspaces. Figures 4 and 5 show the performance comparison between MathNAS and LANA on macrospace MobileNetV3 and microspace NAS-Bench-201 respectively. Accuracy refers to ImageNet validation after 200 independent training epochs. For LANA's assessment, the largest FLOPs network in NAS-Bench-201 serves as the teacher net, with validation accuracy changes noted for each block. As shown in Figure 5(b), while LANA is tailored for macro spaces and struggles in micro spaces, MathNAS exhibits consistent prediction accuracy even in the micro-space NAS-Bench-201, underscoring its search space adaptability.

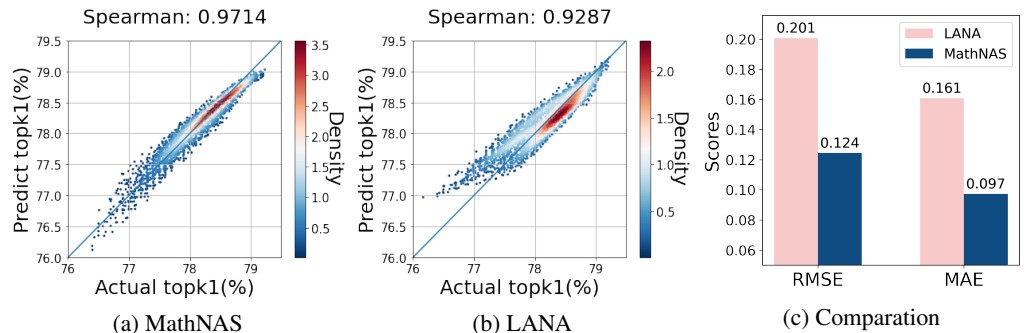

(a) MathNAS        (b) LANA        (c) Comparation

Figure 4: Evaluation of accuracy prediction capabilities of MathNAS versus LANA in the MobileNetV3 (x 1.2) search space on ImageNet.

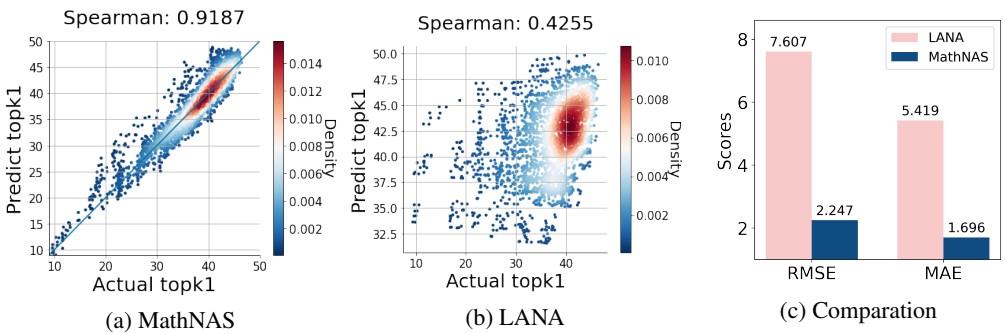

(a) MathNAS        (b) LANA        (c) Comparation

Figure 5: Accuracy prediction comparison between MathNAS and LANA within the NAS-Bench-201 search space.

Overall, MathNAS solves the following limitations compared to previous blockwise methods:

- It does not depend on distillation but proposes a new block evaluation method based on the observed relationship between FLOPs and delta-Accuracy. This method is mathematically efficient and succinct, and it has been theoretically validated across different search spaces.

- It applies to a wider variety of search spaces, beyond the classical ones apart from the macro search space, such as the micro search space of NB201.

- Its evaluation of block performance and network accuracy prediction is more precise.

- It can find more superior architectures based on a full-space search, holding true even when compared to non-blockwise NAS methods. Moreover, the time complexity of the search algorithm is at a polynomial level.

### D.3 Experiment on Dynamic Network

In the main body of this paper, we analyze the dynamic performance of MathNAS and state-of-the-art (SOTA) dynamic networks on real devices. Here, we extend this analysis by providing a more comprehensive comparison of the on-device performance of MathNAS and SOTA dynamic networks.

**Search Time.** We delve into the search time required by MathNAS under various latency constraints, comparing it with other methodologies. Figure 6 (a) reveals that the search time varies under different latency constraints. More specifically, when the latency constraint is either relatively large or small, the search duration decreases. This is primarily due to the pruning process undertaken using the branch and bound method, which improves search efficiency by employing upper and lower bounds. A larger latency constraint results in a higher lower bound, and a smaller latency constraint results in a lower upper bound. This in turn increases the extent of pruning, reducing the search range and thus enhancing the search speed. Moreover, we compare the search times of various NAS methods. As Figure 6 (b) indicates, the search time on Linprog is longer than that on Gurobipy. This is because Gurobipy can utilize all available hardware resources on a device, whereas Linprog relies solely on a single CPU. Despite this, Linprog's search time is about 1/3300 of that of OFA, resulting in a significant reduction in NAS search time. For Gurobipy, the search time is 1/80000 of the OFA search time.

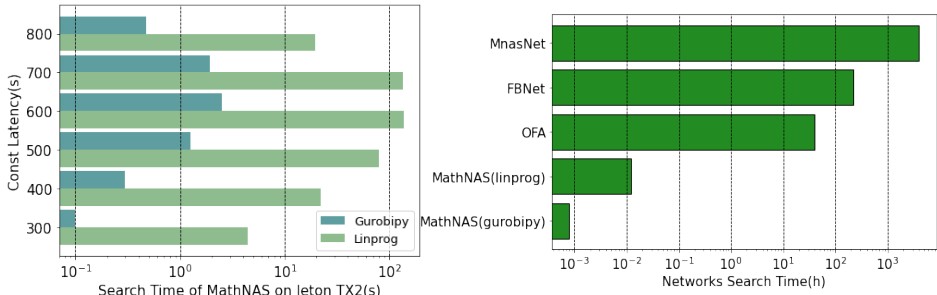

Figure 6: Experiments are performed on OFA search space and Jetson TX2 GPU. (a) Search time of MathNAS under different latency constraints. (b) Comparison of search times of MathNAS with other NAS methods.

**Energy.** The key objective of dynamic networks is to adaptively switch architectures based on the device environment. As such, the time and energy consumed during architecture switching, as well as the ability to respond to environmental changes, are crucial metrics for dynamic network methods. In this section, we scrutinize the performance of MathNAS in dynamic environments with latency and energy as metrics. We also utilize the UNI-T UT658 power monitor to gauge energy consumption during the switching process. Initial findings, as displayed in Figure 7, indicate that both the switching energy and latency of MathNAS are significantly less than those of Dynamic-OFA. This is primarily due to the fact that MathNAS only switches necessary sub-blocks, while Dynamic-OFA switches entire sub-nets. When deployed on Raspberry Pi, MathNAS requires approximately 80% less energy to switch sub-nets compared to Dynamic-OFA. Furthermore, the latency and energy required for MathNAS to switch sub-nets are only marginally higher than those required by AutoSlim, despite the fact that the size of candidate networks in AutoSlim is much smaller than in MathNAS.

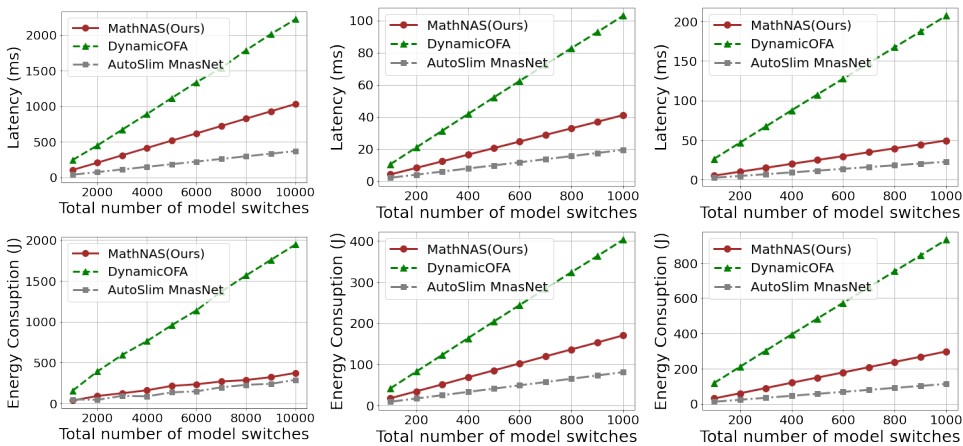

Figure 7: Latency (top) and energy consumption (bottom) required for different dynamic networks switching on Raspberry Pi (left), TX2GPU (middle) and TX2CPU (right).

# E Searched Architecture Visualization

In this section, we provide visualizations of our searched architectures, including MathNAS-MB and MathNAS-T, and dynamic networks.

### E.1 Visualization Architecture on MobileNet-V3

Figure 8 shows the visual architecture of the network searched by MathNAS on the MobileNet-V3 search space.

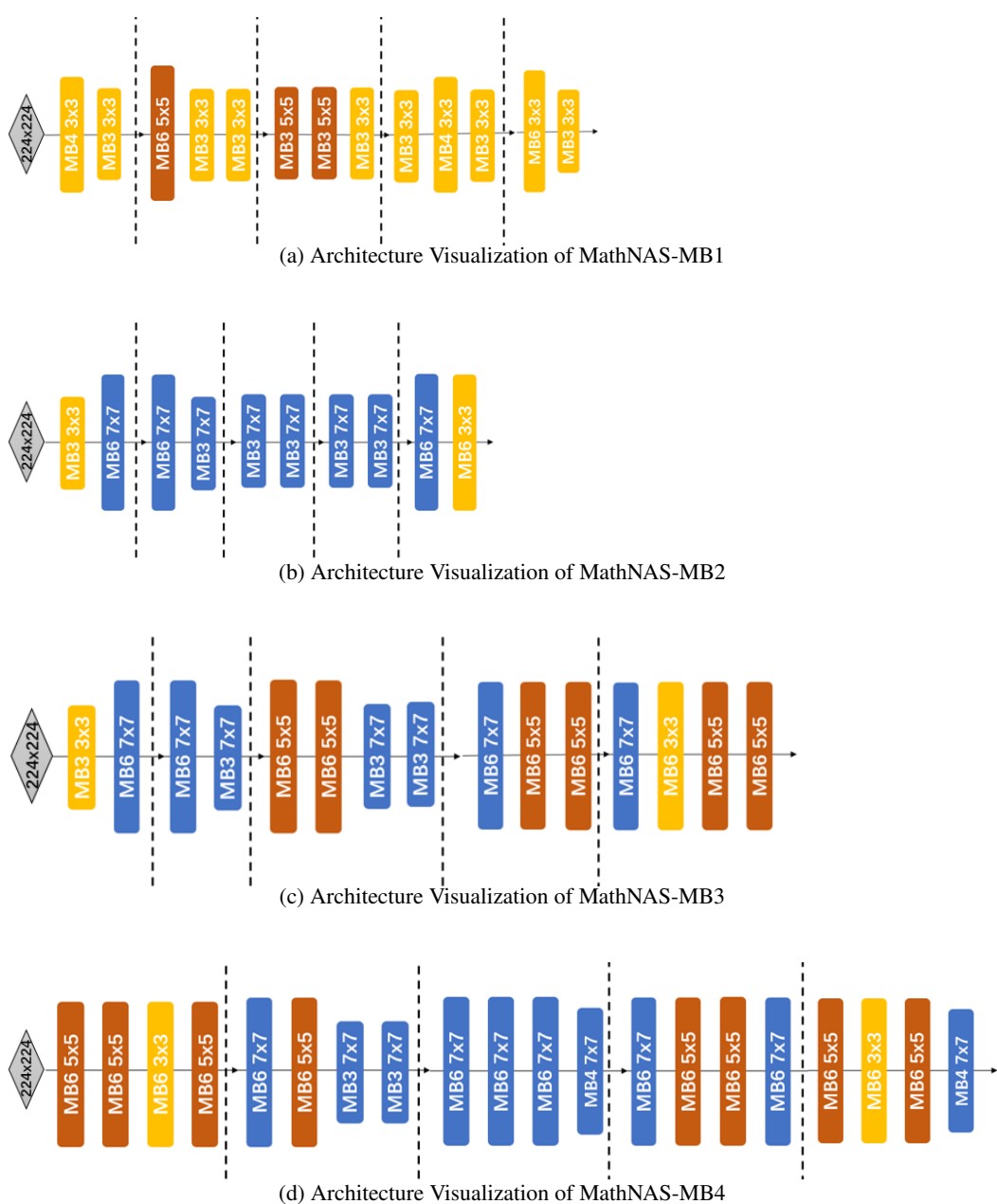

(a) Architecture Visualization of MathNAS-MB1

(b) Architecture Visualization of MathNAS-MB2

(c) Architecture Visualization of MathNAS-MB3

(d) Architecture Visualization of MathNAS-MB4

Figure 8: Architecture Visualization of MathNAS-MB Network.

## E.2 Visualization Architecture on SuperViT Space

Table 7 shows the visual architecture of the network searched by MathNAS on the SuperViT search space.

Table 7: Architecture visualization of MathNAS-T model. 'c' denotes the number of output channels, 'd' denotes the number of layers, 'ks' denotes kernel size, 'e' denotes expansion ratio, 'k' denotes the number of windows, and 's' denotes stride.

| | MathNAS-T1 | MathNAS-T2 | MathNAS-T3 | MathNAS-T4 | MathNAS-T5 |
|---|---|---|---|---|---|
| Conv | c:16
d:1
ks:3
s:2 | c:16
d:1
ks:3
s:2 | c:24
d:1
ks:3
s:2 | c:16
d:1
ks:3
s:2 | c:24
d:1
ks:3
s:2 |
| MBConv-1 | c:16
d:3
ks:3
e:1
s:1 | c:16
d:3
ks:3
e:1
s:1 | c:16
d:1
ks:
e:1
s:1 | c:16
d:1
ks:5
e:1
s:1 | c:24
d:2
ks:3
e:1
s:1 |
| MBConv-2 | c:24
d:3
ks:3
e:4
s:2 | c:24
d:4
ks:3
e:4
s:2 | c:24
d:3
ks:
e:5
s:2 | c:24
d:3
ks:3
e:5
s:2 | c:24
d:4
ks:3
e:5
s:2 |
| MBConv-3 | c:32
d:3
ks:3
e:4
s:2 | c:40
d:3
ks:3
e:4
s:2 | c:40
d:4
ks:
e:4
s:2 | c:32
d:6
ks:3
e:4
s:2 | c:40
d:6
ks:3
e:5
s:2 |
| Transformer-4 | c:64
d:3
k:3
e:1
s:2 | c:64
d:4
k:3
e:1
s:2 | c:64
d:3
k:3
e:1
s:2 | c:72
d:4
k:3
e:1
s:2 | c:72
d:6
k:3
e:1
s:2 |
| Transformer-5 | c:112
d:3
e:1
s:2 | c:112
d:3
e:1
s:2 | c:120
d:6
e:1
s:2 | c:120
d:8
e:1
s:2 | c:128
d:8
e:1
s:2 |
| Transformer-6 | c:176
d:3
e:1
s:1 | c:176
d:3
e:1
s:1 | c:184
d:6
e:1
s:1 | c:176
d:8
e:1
s:1 | c:184
d:8
e:1
s:1 |
| Transformer-7 | c:208
d:3
e:1
s:2 | c:208
d:3
e:1
s:2 | c:216
d:6
e:1
s:2 | c:216
d:5
e:1
s:2 | c:216
d:5
e:1
s:2 |
| MBPool | 1792 | 1792 | 1792 | 1792 | 1792 |
| Resolution | 192 | 224 | 256 | 288 | 288 |

## E.3 Visualization Architecture on Dynamic Network

Figure 9 shows the Pareto-optimal sub-blocks selected on Jetson TX2 CPU/ Raspberry Pi and Jetson TX2 GPU in the MobileNet-v3 search space, respectively. GPU prefers shallow and wide block architectures, thus its sub-blocks have fewer layers but larger kernel size and channel number, while

CPU prefers deep and narrow block architectures, thus its sub-blocks have more layers, but the kernel size and channel number are smaller.

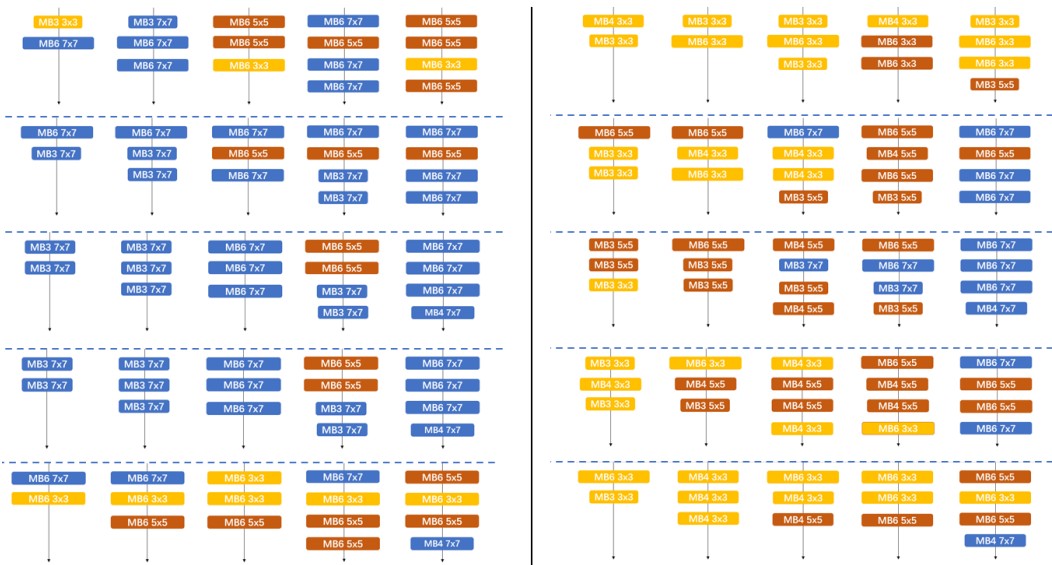

Figure 9: CPU (right) and GPU (left) Pareto-optimal sub-blocks selected in the MobileNet-V3 super-net. From top to bottom is from block1 to block5. Blocks in the same row are sub-blocks of the same super-block, and for the same device, the size of the sub-blocks increases sequentially from left to right.

## F   Discussion

### F.1   Limitations and Future Work

In this section, we discuss the limitations of this work.

- The theoretical explanation of the law of FLOPs and accuracy changes proposed by Math-NAS needs to be strengthened. We intend to continue to explore and try to give a complete theoretical proof from the aspect of network interpretability.
- Zero-shot NAS algorithms have been proven to be more efficient. Our future goal is to investigate the potential of applying MathNAS to zero-shot NAS algorithms.

### F.2   Potential negative societal impact

Our proposed technique for rapidly designing network architectures may lead to unemployment of network architecture designers. The technology can also be misused by those who might create evil artificial intelligence.