# OpenReview forum: "MathNAS: If Blocks Have a Role in Mathematical Architecture Design"
_NeurIPS.cc/2023/Conference — NeurIPS 2023 poster_

### Official Review · Reviewer_qRGY · 2023-06-28

**Soundness:** 3 good
**Presentation:** 3 good
**Contribution:** 3 good
**Rating:** 5
**Confidence:** 4

**Summary:**

The paper introduces MathNAS, a novel approach to Neural Architecture Search (NAS) that utilizes mathematical programming techniques. MathNAS aims to improve the efficiency and accuracy of architecture search by dividing the search space into smaller building blocks and predicting network performance based on the performance of these blocks. The authors validate the effectiveness of MathNAS on computer vision (CV) and natural language processing (NLP) tasks, showcasing its superior performance compared to state-of-the-art models.

**Strengths:**

1. Efficient Performance Evaluation: MathNAS proposes a general framework for evaluating candidate networks by estimating block performance first and then combining them to predict network performance. This approach greatly improves evaluation efficiency .
2. Reduced Search Complexity: By establishing a mapping between block performance and network performance, MathNAS transforms NAS into an Integer Linear Programming (ILP) problem, reducing the search complexity to polynomial time .
3. Superior Results: The authors demonstrate MathNAS's capabilities by considering key performance indices such as accuracy, latency, and energy. The results show that MathNAS outperforms state-of-the-art models in terms of these metrics .

**Weaknesses:**

1. Lack of Detailed Implementation: The paper provides a high-level overview of MathNAS but lacks detailed implementation information. More specific details about the mathematical programming techniques used and the specific algorithms employed would have been beneficial.

2. Section 2.2 of the paper presents some significant ambiguities that complicate understanding of the proposed method:

a. The paper does not clearly explain the definitions of "inherent capability" and "interactive capability". This makes it challenging to understand their intended meanings and how they function within the given neural network context.

b. There is a lack of clarity on how the changes in these capabilities, represented as $\Delta \phi\left(\mathcal{B}{(i, 1) \rightarrow(i, j)}\right)$ and $\Delta \Phi\left(\mathcal{B}{(i, 1) \rightarrow(i, j)}\right)$, are evaluated. The methodology for these calculations needs to be explained more explicitly.

c. Equation 3 lacks clarity due to the aforementioned uncertainties. Without clear definitions and calculation methods for the terms on the right-hand side, it is impossible to meaningfully interpret or apply this equation.

d. Additionally, the processes of module swapping and internal adjustment are not well-defined. More details on how these operations are performed, both in terms of selecting which modules to swap and how the internal adjustments are made post-swap, are needed for a comprehensive understanding and replication of the method. Does every module have the sample input/output shape?


3. Limited Discussion on Generalization: Although the paper mentions MathNAS's remarkable generalization capabilities in designing efficient architectures for NLP tasks, there is limited discussion on the factors contributing to this generalization and how it compares to other NAS methods.

**Questions:**

1. Can you clarify the approach or strategy employed to divide the networks into individual modules or blocks?

2. Based on observations from Figure 1b, the Floating Point Operations Per Second (FLOPs) appear to have minimal impact on latency. Could you provide an explanation or insight into why this phenomenon occurs?

3. Have you considered implementing a Graph Neural Network (GNN) model designed to predict performance variances that may arise as a result of interchanging different modules within the network? What outcomes or implications might this approach yield?

---

> ### Author Rebuttal · Authors · 2023-08-08
>
> Thank you for your thoughtful review of our work.
>
> ---
> # Clarification about Section 2.2
> ## Clarifications on Definition and Calculation of Equation 3
> *The answers here correspond to parts a, b, and c of the original question.*
>
> Both theoretically and empirically, we have shown that the effects of a specific block switch differs across various architectures. Delving into these discrepancies, we define the calculation of block performance as:
> $$b^{A}\_{i,j}=\overline{\Delta Acc\left(\mathcal{N}^{\Omega}\_{(i,1) \rightarrow  (i,j)}\right )}$$.
> **This formula is designated for the computation of the performance of $b\_{i,j}$**.
>
> Furthermore, we propose that the effect of block performance on network accuracy should be approached from two angles: inherent capability and interactive capability. These two concepts help illuminate the meaning behind our block performance formula (**though it's important to note they serve explanatory purposes and aren't directly employed in the calculation**). Hence, we have:
> $$\overline{\Delta Acc\left(\mathcal{N}^{\Omega}\_{(i,1) \rightarrow  (i,j)}\right )} = \Delta\phi(\mathcal{B}\_{(i,1)\rightarrow (i,j)}) + \Delta\Phi(\mathcal{B}\_{(i,1)\rightarrow (i,j)})$$.
> Wherein:
> - The term *inherent capability* refers to the intrinsic ability of the block.
> - The term *interactive capability* denotes the capacity of a block when considering its influence on other blocks within the network.
>
> To draw an analogy, consider a team where an individual's performance is not just a reflection of their inherent skill but also the influence they exert on their teammates. Similarly, a network can be viewed as a team composed of blocks.
>
> ## Details on Module Switching and Internal Adjustments
> *The answers here correspond to part d of the original question.*
>
> For all replacement blocks within the search space, a performance evaluation is required. Specifically, to evaluate the performance of block $b\_{ij}$, we consider the switch from $b\_{i1}$ to $b\_{ij}$.
>
> After a block swap, there's no need for internal adjustments. This is because for all candidate blocks contained within the $i$-th block node, their input and output tensor spatial dimensions match those in the search space.
>
> ---
> # Other Questions
> ## Detailed implementation:
> In our *common response*, we've provided detailed technical specifics regarding the solution to the ILP equation. These technical details will be supplemented in our revised version.
>
> ## Discussion of generalization on NLP tasks:
> Here, we'd like to elucidate the generalization capability of MathNAS across diverse tasks.
>
> As mentioned in the paper, MathNAS is applicable to various block architectures, including convolutional blocks and transformer blocks.
> **The architectural applicability of MathNAS is task-agnostic.**
> This suggests that, for a range of tasks spanning CV and NLP, MathNAS is theoretically applicable as long as the search space can be structured in a block format.
>
> Moreover, the Transformer architecture has demonstrated significant potential in NLP tasks.
> Experiments further revealed that MathNAS exhibits exceptional performance within the search spaces of Transformer architectures, including SuperViT and SuperTransformer.
> This, to some extent, highlights the commendable task generalization of MathNAS on NLP tasks.
>
> ## Division of blocks:
> A detailed description of the block division for different search spaces can be found in the Appendix D.1.
>
> Here, we provide a brief overview of our block partitioning strategy:
>
> - For the **MobileNetV3** search space, we adhere to the original block division, considering one inverted residual structure as a single block.
> - For the **SuperTransformer** search space, we classify two encoder layers as one block and one decoder layer as another block.
> - For the **SuperViT** search space, we again follow the original block categorization, defining either a CNN or a Transformer block as a single block.
>  - For the **NAS-Bench-201** search space, our concept of a block corresponds to an edge operation in a GNN, where each operation on an edge equates to a block.
>
> ## Interpretation of latency experimental results:
> Figure 1 illustrates the impact of FLOPs on network performance (accuracy, latency, energy) during block switches. In other words, it reveals the contribution of blocks to the performance of networks with varying FLOPs.
>
> The minimal variations in latency shown in Figure 1b suggest that **the contributions of different blocks to the network's latency are independent**, which aligns with our understanding. Taking a network composed of sequentially connected blocks as an example, the network's required latency is roughly the sum of the independent latencies needed for each block.
>
> ## Implementation on the GNN model:
>
> In fact, we believe that our experimental verification of MathNAS on the NAS-Bench-201 search space can serve as a testament to its compatibility with GNN networks.
>
> NAS-Bench-201 represents a micro search space, consisting of five identical GNN blocks where the edges of a GNN can select different operations.
>
> In the context of the paper, the block switch operation, when applied to NAS-Bench-201, essentially refers to changing the operation on a particular edge while keeping the operations on other edges constant. This effect mirrors the application of MathNAS to a complete GNN model.
> Hence, from the validation of MathNAS on NAS-Bench-201, we can infer its suitability for GNN networks.
>
> We will clarify this point in our revised version.

---

> ### Author Response · Authors · 2023-08-21
> **Anticipating Your Feedback**
>
> With the rebuttal deadline fast approaching, we kindly request your feedback. If you have any questions or concerns, please don't hesitate to reach out to us.

---

### Official Review · Reviewer_HYbp · 2023-07-07

**Soundness:** 3 good
**Presentation:** 2 fair
**Contribution:** 4 excellent
**Rating:** 7
**Confidence:** 4

**Summary:**

The paper proposes a way to perform neural architecture search (NAS) by solving an integer program to identify the constituents of a block-modular neural net, having pre-computed estimations of performance for the separate blocks by a novel method. The resulting "MathNAS" approach is demonstrated in various experiments to be able to yield competitive neural network designs compared to other state of the art NAS methods.

-- update: I have read and acknowledged all other reviews and the authors rebuttals, see discussion. --

**Strengths:**

The novel performance estimation method appears to be well-motivated and leads to the possibility of applying mixed-integer programming to compute an optimal architecture (w.r.t. the utilized performance estimates). The numerical results indicate that this works well in practice, giving somewhat better or comparable results with notable savings in training time. This new NAS pipeline also shows promise for future refinements, e.g., regarding the MIP problems used for the search once performance estimates have been pre-computed, given the inherent flexibility of MIP to model a variety of different aspects.

**Weaknesses:**

The paper must be thoroughly proofread for English language and grammar/typos. The supplementary document is as long as the main paper, but provides mostly only some small additional information; some things I hoped to find in the supplement were however not included, e.g., an explanation for the MIP objective function and a description or at least literature reference for how to turn the fractional objective into a linear one. Math typesetting especially of optimization programs can be improved. Also, I got the impression that the authors are trying a bit too hard to "sell" their ideas; maybe tone down the language a bit (there's very many usages of words like "impressive" or "remarkable" and similar phrasings praising the paper's results). Finally, there are several parts that remained somewhat unclear to me (see "Questions"), so the presentation can also be improved in terms of clarity/rigor.

**Questions:**

- what exactly do you mean by "polynomial-time search complexity"? It is quite confusing to read something like "...can be transformed into an integer programming problem, which further reduces the search complexity to polynomial time" -- solving IPs is generally not possible in polynomial time!   I suspect the authors mean that they need to pre-compute polynomially many values (for their performance estimation scheme), but this does not really become clear.
- In the same vein, it is mentioned that some neural network training is necessary, but it is never clearly stated what and when. Again, I guess training is required to pre-compute the block performance scores somehow, but the details of this are too vague; this certainly needs to be clarified.
- Could you describe a bit more how flexible the block/modular architecture is? (i.e., mention earlier that blocks can be construed of essentially any (finite) number of sub-network; this is never clearly stated and only becomes more clearly apparent in the supplementary document where details on the search spaces in the different experiments are given).
- You claim at several points that the integer programs were solved with Gurobi (that is the solver name; Gurobipy is merely its Python interface...) on a GPU.  I am quite sure that Gurobi does not run on GPUs. MIP solvers currently do not benefit from GPU vs. CPU computations; Gurobi even explain this on their website. So, please clarify and correct/elaborate in the paper! (Also, state the Gurobi version you used.)
- Please clarify what exactly is being sampled (and to what purpose exactly) in Section 2.3
- Please justify the IP objective function and clarify/give a reference for how the fractional IP can be turned into a linear IP.

**Limitations:**

Limitations have by and large been addressed sufficiently.

---

> ### Author Rebuttal · Authors · 2023-08-08
>
> Thank you for your thoughtful review of our work.
>
> ---
> # Polynomial-time Search Complexity
>
> As you surmised, the polynomial time referenced here does not pertain to the duration required to solve the ILP problem. Instead, **it denotes the time complexity of the MathNAS algorithm in identifying the optimal architecture, or more precisely, the number of networks that need evaluation within the search space**. Only after evaluating the requisite networks do we proceed to solve the ILP problem. Similar forms of time complexity descriptions have appeared in other MP-NAS works, such as in LayerNAS [1].
>
> [1] Yicheng Fan et al. LayerNAS: Neural Architecture Search in Polynomial Complexity. 2023.
>
> ---
> # Network Training
> Here, we clarify the network training aspects within the MathNAS algorithm.
>
> MathNAS operates in two stages: an offline block evaluation stage and an online search stage. In the offline stage, we first assess the accuracy of $m\* n$ sampled networks and then calculate block performance based on these evaluations.
>
> **[WHEN]** This network evaluation stage (offline stage) indeed encompasses the network training process.
>
> **[WHAT]** Specifically, for our experiments in the MobileNetV3 [2], SuperViT [3] and SuperTransformer [4] search spaces, we utilize networks pretrained from prior work. For the NAS-Bench-201 [5] search space, we rely on the accuracy on the validation set of networks independently trained for 200 epochs. In the revised version, we will elaborate on these offline costs and incorporate them into the overall cost of the MathNAS algorithm.
>
> [2] Andrew Howard et al. Searching for mobilenetv3. 2019.
>
> [3] Chengyue Gong et al. Nasvit: Neural architecture search for efficient vision transformers with gradient conflict aware supernet training. 2021
>
> [4] Hanrui Wang et al. Hat: Hardware-aware transformers for efficient natural language processing. 2020.
>
> [5] Xuanyi Dong et al. Nas-bench-201: Extending the scope of reproducible neural architecture search. 2020.
>
> ---
> # Block Architecture Flexibility
> Here, we provide a detailed description of block architecture flexibility.
>
> Given a search space comprised of $m$ block nodes, each block node has $n$ candidate blocks. These blocks can be of any neural network type, including CNNs and Transformers, with no structural constraints. Additionally, a candidate block can be a 'skip', indicating variability in the number of network blocks.
>
> For instance, in the case of an inverted residual block, one can choose different numbers of network layers, kernel sizes, and expansion ratios. For a transformer block, one can select varying numbers of encoder/decoder layers, hidden dimensions, and the number of attention module heads.
>
> **As long as the spatial dimensions of the input and output tensors of the candidate blocks match the dimensions at the corresponding positions in the search space, the MathNAS algorithm can effectively conduct the search.**
>
> ---
> # Sampling strategy in Section 2.3
> To compute the accuracy of block $b\_{i,j}$, we must evaluate the mean change in network accuracy when all networks in the search space containing block $b\_{i,1}$ undergo a specific block replacement process - switching from $b\_{i,1}$ to $b\_{i,j}$ (ref: Equation 3).
>
> This implies that in order to calculate $b\_{i,j}^A$, we need to determine the accuracy for $m^{(n-1)}$ networks. However, given the observed inverse relationship between network FLOPs and delta-Accuracy, we can estimate $b\_{i,j}^A$ based on the accuracy variation of a network with average FLOPs (ref: paper 2.3 Single Network Sampling Strategy).
>
> **In summary, to reduce algorithmic complexity, we sample a specific network from $\mathcal{N}^{\Omega}\_{(i,j)}$ ( i.e., all  $m^{(n-1)}$  networks containing $b\_{i,j}$ ) whose FLOPs equal the average FLOPs of $\mathcal{N}^{\Omega}\_{(i,j)}$.**
>
> We use the change in accuracy of this sampled network as a proxy for the average accuracy change of all networks, thereby deriving the accuracy for block $b\_{i,j}$. Experimental results show this sampling approach introduces minimal error.
>
> ---
> # Other Questions
> ## Writing Problem:
> Thanks for your patient and meticulous review.
> We have rechecked and corrected the grammar and spelling throughout the paper.
> Additionally, we've reorganized the Appendix and provided corresponding references in the paper.
> We have also considered improvements in the formatting of mathematical formulas and refined our choice of words.
> These writing issues will be addressed in the revised version.
>
> ## Use of Gurobi:
> Upon re-examining our code related to the dynamic network experiments, we discovered that when running MathNAS on the TX2 GPU, the equations were actually being solved on the CPU using the Gurobipy.
> Subsequently, the solved architecture was loaded onto the GPU for execution. Hence, the reported search time primarily reflects the equation-solving time on the TX2 CPU, which explains why the reported CPU and GPU solution times are identical. We apologize for any confusion our initial description may have caused and will clarify this matter in the revised version.
>
> ## ILP objective function:
> In our *common response*, we've provided the technical details of how to convert to an ILP problem and solve it. We will add these technical details in our revised version.

---

> > ### Comment · Reviewer_HYbp · 2023-08-10
> >
> > I thank the authors for their explanations and additional information in response to my own and the other reviews. All issues seem to have been addressed appropriately, so I uphold my opinion that this paper should be accepted.

---

### Official Review · Reviewer_qYEV · 2023-07-07

**Soundness:** 3 good
**Presentation:** 3 good
**Contribution:** 3 good
**Rating:** 7
**Confidence:** 3

**Summary:**

This paper introduces MathNAS, a mathematical programming NAS algorithm. MathNAS maps network performance to block performance, enabling the prediction of accuracy, latency, and energy consumption of large networks based on their constituent modules. This algorithm reduces the search space from n^m to m*n, allowing the NAS problem to be solved as an Integer Linear Programming problem.

**Strengths:**

Overall, this paper is well-written and organized, making it easy to follow along. The authors did a great job explaining their methods in detail, complete with loads of equations and clear explanations. Plus, the proposed algorithm has undergone numerous experiments under different models and application fields, which consistently showcase its effectiveness.

**Weaknesses:**

1)	In all the experiments, MathNAS takes significantly less "search time" than the other methods, especially some methods need thousands of hours while MathNAS only requires several seconds, which is very impressive. Is the time for all the other methods include the model training time (either training time of supernet or the subnet)? If yes, for the fair comparison, for the MathNAS, should the time that needed to train the base model be added to this time as well? Also, as algorithm 1 shows that there are three steps for MathNAS to determine the best architecture, should the time that calculate each block's performance also be added to the total "search time"?

2)	The order of the tables and the description of the experimental results in the paper do not follow the same order. For example, the text follows GNN -> CNNs (Table 2) -> ViT -> NLP (Table 1) -> Dynamic Networks (Table 3), which leads is hard to follow.

3)	A typo in paragraph "MathNAS for Mobile CNNs ...", the top-1 accuracy of MathNAS-MB1 is 75.9% as shown in the table, not 76.4%.


**Questions:**

In all the experiments, MathNAS takes significantly less "search time" than the other methods, especially some methods need thousands of hours while MathNAS only requires several seconds, which is very impressive. Is the time for all the other methods include the model training time (either training time of supernet or the subnet)? If yes, for the fair comparison, for the MathNAS, should the time that needed to train the base model be added to this time as well? Also, as algorithm 1 shows that there are three steps for MathNAS to determine the best architecture, should the time that calculate each block's performance also be added to the total "search time"?

---

> ### Author Rebuttal · Authors · 2023-08-08
>
> Thank you for your thoughtful review of our work. Here we respond point-to-point to your questions.
>
> ---
> # Search Time
> We explain the search times mentioned in the paper in the *common response*.
>
> To ensure a fair comparison, as you rightly pointed out, we will include the training time for the MathNAS base model in the revised version.
>
> ---
> # Description Order
> We are aware of this issue and will address and rectify it in the revised version. We sincerely apologize for any inconvenience caused during your reading.
>
> ---
> # Misspell
> Once again, we greatly appreciate your patient and meticulous review.
>
> Indeed, this oversight was our mistake. We have rechecked the number and will incorporate the necessary corrections during our revision.
>
> Accurate data presentation is vital for the advancement of NAS research. To ensure our data is correctly represented, we have reviewed all other data points in addition to the ones you highlighted.

---

### Official Review · Reviewer_8XzK · 2023-07-12

**Soundness:** 3 good
**Presentation:** 3 good
**Contribution:** 3 good
**Rating:** 6
**Confidence:** 4

**Summary:**

This paper introduces MathNAS, a blockwise NAS framework. It begins by estimating the performance of the building blocks within the search space and then predicts the overall performance of a network based on the performance of its individual building blocks. This approach effectively reduces the complexity of network search from O(n^m) to O(nm). Additionally, a block performance evaluation scheme, utilizing average FLOPs, is proposed to further reduce the time complexity from O(n^m) to O(nm). Lastly, the network search process is formulated as an Integer Linear Programming (ILP) problem.

**Strengths:**

* Clarity: The paper is effectively written, employing appropriate mathematical notations throughout.

* Quality: The experiments conducted in the paper are of high quality, as they include a comprehensive comparison with well-known networks in various domains, such as tabular search space, mobile CNN, ViT, and NLP.

* Originality and significance: While the paper presents valuable contributions, there are some concerns that should be addressed. Please refer to the weaknesses section outlined below.

**Weaknesses:**

The paper does not cite a few highly relevant papers on blockwise NAS.
* LANA: Latency Aware Network Acceleration. https://arxiv.org/pdf/2107.10624.pdf
* Distilling Optimal Neural Networks: Rapid Search in Diverse Spaces. https://arxiv.org/pdf/2012.08859.pdf
* BLOX: Macro Neural Architecture Search Benchmark and Algorithms. https://arxiv.org/pdf/2210.07271.pdf

The authors should thoroughly discuss the distinctions between their proposed approach and the previously mentioned works, while also considering the possibility of conducting experiments to compare them. Specifically, LANA adopts a constrained ILP approach that relies on block metrics derived from factors such as delta-accuracy and delta-latency. The proposed approach bears resemblance to LANA.

While the authors have made an effort to explain the validity of their approach, as depicted in Figure 1 and Section C of the appendix, it remains unconvincing that Equation 2 holds true under all circumstances. Additionally, Equations 3 and 4 are similar to those presented in LANA.

**Questions:**

* Figure 1 in the appendix requires clarification. It appears to suggest the presence of three levels of blocks, with each level accommodating parallel blocks. Could you please clarify whether the blocks are connected sequentially or in parallel? Additionally, how many levels of blocks are there?

* In Algorithm 1, there is an inconsistency with the existence of b_{i,0}, which does not align with the range specified for i (i={1, m}). Is this a typographical error?

* Regarding the results on NB-201, it would be beneficial to know if multiple runs were performed to obtain averages using different seeds. Similarly, for the Mobile CNN, ViT, and NLP networks, were multiple runs conducted to calculate the means and variances?


**Limitations:**

The paper does not explicitly mention the limitations and potential negative social impact of their work.

---

> ### Author Rebuttal · Authors · 2023-08-08
>
> Thank you for your thoughtful review of our work.
>
> ---
> # Originality and Contribution
> #### Regarding the comparison of the effectiveness of MathNAS and blockwise methods, we responded in the *common response*. At the same time, we supplemented the experimental comparison between MathNAS and the block-wise methods, refer to *rebuttle PDF*.
> Here we mainly clarify the difference between our work and LANA.
>
> ### Block Performance Evaluation
>
> **LANA** switches blocks on the teacher net (typically the largest network within the search space) and directly uses the resultant accuracy change as the block's performance metric.
>
> **MathNAS** conducts block switches on a network whose FLOPs represent the average within the search space. This particular network is chosen based on the observed inverse relationship between FLOPs and delta-Accuracy. We estimate block performance using the accuracy changes observed on this network. Experimental results indicate that this is a more efficient and accurate method for evaluating block performance.
>
> ### Network Accuracy Evaluation
>
> **LANA** estimates network performance by simply summing up the block performance. It assumes that the impact of a specific block switch remains consistent across different architectures.
>
> **MathNAS** employs a weighted evaluation of block performances based on the network's FLOPs, thereby offering a more precise assessment of each block's influence on overall network accuracy. This method subsequently aids in more accurate network performance predictions. Furthermore, both theoretically and empirically, we've demonstrated that the effects of a specific block switch vary across different architectures. While exploring these variances, we suggest that the influence of block performance on network accuracy can be explored from two perspectives: inherent capability and interactive capability.
>
> ### Network Latency/Energy Consumption Evaluation
>
> **LANA** estimates network latency by summing up block latencies, which restricts LANA's prediction method to be suitable only for macro search spaces with sequential block connections.
>
> **MathNAS** calculates the delta-Latency for each block based on the obtained block latency. It considers the network latency to be equal to the difference between the latency of the initial architecture and the delta-latency of each block comprising the network. This makes our latency prediction method applicable to micro search spaces, such as NAS-Bench-201.
>
> #### LANA and MathNAS both employ the concept of delta-Accuracy for estimating network performance. This similarity might lead one to believe that Equations 3 and 4 in MathNAS resemble those in LANA. However, it's crucial to note that the block performance evaluation method and the network performance prediction formula in MathNAS are fundamentally distinct from those in LANA.
>
> ---
> # Other Questions
>
> ## Figure 1 in the appendix:
>
> **In fact, this figure does not represent a specific search space.** Our intention is to demonstrate the architectural versatility of MathNAS, that is, it can be applied to search spaces where the block structure is serial or parallel. Specifically, we used four classic search spaces during the experiment: NASBench201, MobileNetV3, Supervit and Supertransformer. A detailed description of their architecture is in Appendix D.1. We are aware of the possible misinterpretations that our diagrams can cause. In the latest revised version, we will remove this figure.
>
> ##  Typographical error in Algorithm 1:
>
> Thank you for your meticulous review. Indeed, **this is a typographical error**. In the revised version, we will make the necessary correction.
>
> ## Obtain averages using different seeds:
>
> Given that the essence of our search algorithm is to solve an ILP (Integer Linear Programming) equation, we employed the Gurobipy and Linprog libraries in Python for the solution. Within a fixed time frame, the solution obtained by the solver is unique and deterministic; hence, **there was no need for multiple computations to obtain average values and variance** in our experiments. Details on solving the ILP equation can be found in the "common response" section.
>
> ## Limitations and negative social impact discussion:
>
> In the revised article, we will add a discussion of the limitations of our work and potential negative societal impacts. Here we give a brief overview of it:
>
> ### limitations and future work:
> - The theoretical explanation of the law of FLOPs and accuracy changes proposed by MathNAS needs to be strengthened. We intend to continue to explore and try to give a complete theoretical proof from the aspect of network interpretability.
> - Zero-shot NAS algorithms have been proven to be more efficient. Our future goal is to investigate the potential of applying MathNAS to zero-shot NAS algorithms.
>
> ### Potential negative societal impact:
>
> Our proposed technique for rapidly designing network architectures may lead to unemployment of network architecture designers. The technology can also be misused by those who might create evil artificial intelligence.

---

> > ### Comment · Reviewer_8XzK · 2023-08-16
> >
> > The authors have put in a significant amount of effort to respond to my concerns. It has clarified the difference with prior works. Please make sure the new materials and citations will be included in the revision.
> > Based on the response, I have updated my score.

---

### Official Review · Reviewer_3m52 · 2023-07-23

**Soundness:** 3 good
**Presentation:** 3 good
**Contribution:** 3 good
**Rating:** 6
**Confidence:** 4

**Summary:**

This paper presents a divide-and-conquer neural architecture search methodology that estimates DNN performance based on the individual performance of each block within a DNN. Accuracy is estimated based on the perturbed accuracy of replacing a block within the NN, and latency/energy are also used within a linear model to predict NN characteristics.

**Strengths:**

The paper is generally well-written and the results are strong compared to the chosen baselines from both CNNs and transformer based vision models. The efficacy of the method on 4 search spaces is impressive.

**Weaknesses:**

There are 3 main weeknesses / questions that I have regarding the paper:

1. As a blockwise NAS approach, this paper largely ignores much of the prior literature on the topic, for example, [1,2,3]. These prior works also rely on blockwise layer statistics (sometimes obtained through distillation) then the layerwise performance is assembled either with a learned predictor [1,2] or ILP [3] similar to what is proposed in this work.
2. A large part of the evaluation is done on NB201 which isn't a macro search space. I am a bit confused as to how that evaluation on NB201 demonstrates the presented blockwise methodology -- all blocks within NB201 are identical, just sized differently. Perhaps this is something the authors can clarify in the rebuttal? I understand that the plots in Fig. 1 are showing the accuracy delta when one block is replaced with another, bit this would switch out the entire network basically, right?
3. Some conclusions (e.g. line 156) may be too specific to NB201. The inverse correlation of flops and accuracy is not a hard rule and many counterexamples can be presented. On NB201 specifically, it is widely known that this holds.
4. Search cost is written as "10 minutes" does this include all the time to evaluate individual blocks and build up the model? I think this cost should be quantified clearly in the paper.


[1] Distilling Optimal Neural Networks: Rapid Search in Diverse Spaces
[2] Blockwisely Supervised Neural Architecture Search with Knowledge Distillation
[3] LANA: Latency Aware Network Acceleration
[4] BLOX: Macro Neural Architecture Search Benchmark and Algorithms

**Questions:**

see above

**Limitations:**

see above

---

> ### Author Rebuttal · Authors · 2023-08-08
>
> Thank you for your thoughtful review of our work.
>
> ---
> # Citations
> We have meticulously reviewed the mentioned works and discussed the distinctions between MathNAS and them in the *common response*. Furthermore, we compared the accuracy prediction performance of MathNAS with these works. The experimental results can be found in the *rebuttal pdf*.
>
> ---
> # Experiments on NAS-Bench-201
> First, let's clarify the specific application of MathNAS in the NAS-Bench-201 search space. As you rightly pointed out, NAS-Bench-201 is not a macro search space, and during a block switch, all blocks undergo changes. In this paper, the "block" concept we mentioned in the paper corresponds to the "edge operations" in NAS-Bench-201. Consequently, in Figure 1, the block switch operation within this space refers to the delta accuracy resulting from a change in one edge operation while keeping other edge operations constant. This is consistent with the essential nature of switching in MathNAS.
>
> In the paper, since we also considered macro search spaces beyond NAS-Bench-201, we adopted the "block" concept for modeling and derivation for the sake of uniform representation. NAS-Bench-201 can be viewed as a special case of the MathNAS model.
>
> In the revised version, we will elaborate and clarify this matter further.
>
> ---
> # Generality of Conclusions
> We concur with your observation that conclusions drawn from prior works in the NAS field concerning NAS-Bench-201 may not necessarily hold in other search spaces. However, we believe our experiments demonstrate the generalizability of the proposed rule from two aspects:
> - First, we validated the inverse proportionality rule between delta-Accuracy and FLOPs on both MobileNetV3 and NAS-Bench-201, as depicted in Figure 1 of the paper. MobileNetV3, being a macro search space constructed from sequentially connected inverted residual blocks, contrasts starkly with NAS-Bench-201. The verification in the MobileNetV3 search space underscores the universal applicability of the stated rule across diverse architectures.
> - Second, we tested the network performance prediction formula derived from the inverse relationship between FLOPs and delta-Accuracy (i.e., Equation 5) across four distinct search spaces: NAS-Bench-201, MobileNetV3, SuperViT, and SuperTransformer, as showcased in Figure 3 of the paper. These spaces encompass varied network architectures (detailed descriptions of search space structures can be found in Appendix D.1), including CNN, GCN, and Transformer, representing the majority of contemporary NAS search space architectures. Hence, the validation across these four spaces suggests that the introduced inverse rule and the network performance prediction formula can be applied broadly across most search spaces.
>
> Given the above, we feel that the statement, "A large part of the evaluation is done on NB201," doesn't quite align with our experimental setup. We referenced NAS-Bench-201 more frequently in our paper because it offers detailed and reliable performance information for its contained architectures, which might have inadvertently led to misunderstandings.
>
> ---
> # Search Cost
> After a thorough review, we couldn't locate any mention of a "10 minutes" search cost. Perhaps you are referring to the "10 seconds" mentioned in Table 1.
>
> In the *common response*, we provided an explanation of the search cost. In the revised version, we will provide a clearer elucidation of this aspect.

---

> > ### Comment · Reviewer_3m52 · 2023-08-18
> >
> > The rebuttal text and additional experiments have addressed most of my comments. Comparisons to other blockwise approaches would be a great addition to the paper. Citing and comparing the method to DONNA, DNA, BLOX, LANA is also very important in my opinion.
> >
> > I still don't fully understand how a perturbation of an edge operation in NAS-Bench-201 can be considered a block swap? This changes the whole model (all 9 cells in the case of NAS-Bench-201) and not just 1 block/cell since this is a micro search space. Because most evaluations in this paper are based on this dataset, I think this is an important thing to clearly explain and flesh out. Can you please explain more on this point?
> >
> > I will raise my score by one point assuming I receive an adequate response to the above concern. Thanks for the rest of your work on this rebuttal.

---

> > > ### Author Response · Authors · 2023-08-19
> > >
> > > Thanks for your kind comment.  In the revised version, we will include the comparison between MathNAS and other block-wise methods.
> > >
> > > ---
> > >
> > > To address your concerns, here we clarify the rationale and methodology behind treating perturbations in edge operations within NAS-Bench-201 as block swaps from two perspectives:
> > >
> > > * Practical Implementation (How): **During experiments within the NAS-Bench-201 space, we identify a set of edges in the same position across multiple GNN cells as a single "block".** Given that the structure of GNN cells in the network remains consistent, our focus is on a single cell. Hence, any alteration in an edge operation essentially translates to a corresponding change in all cells, while other edge operations remain static.
> > >
> > > * Theoretical Framework (Why): Regardless of the distinction between macro and micro search spaces, the networks in both are assembled from multiple mutable modules, be it blocks or edges. The capability of the entire network can be represented by the capabilities of these individual modules. In MathNAS, to explore the contribution of module capabilities to the network's performance, we evaluated changes in inherent module capabilities and their interactive capacities during module switches (i.e., Equation 3). **This module evaluation methodology is applicable to our definition of blocks in NAS-Bench-201: alterations in edge operations impact not only the specific edges' output data (inherent capability) but also influence the input and output data of other edges within the network (interactive capability).** Empirical evidence from macro search spaces (MobileNetV3, SuperViT, and SuperTransformer) and micro search space (NAS-Bench-201) also underscores the universality of our block definition and its adaptability across various search spaces.
> > >
> > > We will clarify this matter in our revised version. Please do not hesitate to reach out with any additional queries.

---

### Author Rebuttal · Authors · 2023-08-08

Thank you all for your thoughtful reviews. Here, we respond to some common concerns across all reviewers.

---
# Paper Citations, Novelty, and Supplementary Experiments
Existing blockwise methods such as DNA [1], DONNA [2], and LANA [3] use block distillation techniques to block the teacher model, obtaining the architecture of blocks to be replaced and their performance evaluation. Following this, they use the performance of each block to guide the algorithm in finding models with superior performance.

Recent work [4] has pointed out the limitations of such methods. They depend on an excellent teacher network due to their use of distillation techniques. Furthermore, previous block performance estimation methods are unable to effectively predict actual block performance. Additionally, these methods are only suitable for the macro search space. MathNAS overcomes the above limitations:
- It does not depend on distillation but proposes a new block evaluation method based on the observed relationship between FLOPs and delta-Accuracy. This method is mathematically efficient and succinct, and it has been theoretically validated across different search spaces.
- It applies to a wider variety of search spaces, beyond the classical ones apart from the macro search space, such as the micro search space of NB201 (ref: rebuttal pdf).
- Its evaluation of block performance and network accuracy prediction is more precise (ref: rebuttal pdf).
- It can find more superior architectures based on a full-space search, holding true even when compared to non-blockwise NAS methods. Moreover, the time complexity of the search algorithm is at a polynomial level.

*We have conducted supplementary experiments to compare the applicability of MathNAS and previous blockwise methods across different search spaces, as well as the efficacy of network accuracy predictions. Please refer to the attached pdf for details.*

In the revised version, we will include these references, discuss the differences, and provide supplementary experimental validation.

[1] Changlin Li et al. Blockwisely supervised neural architecture search with knowledge distillation. 2020.

[2] Bert Moons et al. Distilling optimal neural networks: Rapid search in diverse spaces. 2021.

[3] Pavlo Molchanov et al. LANA: latency aware network acceleration. 2022.

[4] Thomas Chau et al. BLOX: Macro Neural Architecture Search Benchmark and Algorithms. 2022.

---
# Search Cost
The search cost of MathNAS consists of two stages: offline network pre-training (conducted only once) and online real-time search.
- During the offline network pre-training, MathNAS evaluates block performance once.
- During online searching, MathNAS is capable of multiple real-time searches based on the current hardware resource constraints.

To negate the influence of variations in GPU models and versions on the pre-training time, and to facilitate comparisons by future researchers, we have adopted pre-trained networks provided by existing works (MobileNet [2], SuperViT [3], SuperTransformer [4], NAS-Bench-201[5]). Consequently, all mentions of MathNAS's search cost in the paper refer solely to the real-time search time on edge devices (i.e., the time taken to solve the ILP problem).

In the revised version, we will consider adding the offline network pre-training time and will adjust the existing statements accordingly for greater clarity.

[5] Andrew Howard et al. Searching for mobilenetv3. 2019.

[6] Chengyue Gong et al. Nasvit: Neural architecture search for efficient vision transformers with gradient conflict aware supernet training. 2021

[7] Hanrui Wang et al. Hat: Hardware-aware transformers for efficient natural language processing. 2020.

[8] Xuanyi Dong et al. Nas-bench-201: Extending the scope of reproducible neural architecture search. 2020.

---
# Searching Equation Solving Details
In this section, we describe in detail the solution of the fractional objective function programming equation proposed in the paper.
The solution is divided into two steps.
1. Convert the original equation into an integer linear programming equation.
2. Solve the ILP equation.

### Equation Transformation
In order to transform the equation into an ILP problem, we first perform variable substitution on the original equation [9].
$$
\text{let}\quad b^{\widetilde{B}}\_{i,j}=\cfrac{b^B\_{i,j}}{\sum\_{i=1}^m\sum\_{i=1}^nb^F\_{i,j}*b^B\_{i,j}},\quad z=\cfrac{1}{\sum\_{i=1}^m\sum\_{i=1}^nb^F\_{i,j}*b^B\_{i,j}}
$$
Then the original equation can be transformed into the following integer linear programming problem:

$$
\begin{split}
     &O = \min\limits\_{b^{\widetilde{B}},z}{ (\sum\_{i=1}^m\sum\_{j=1}^nb^A\_{i,j}\*b^{\widetilde{B}}\_{i,j}\*\overline{\mathcal{F}(\mathcal{N})})} \\\\
     &s.t. \\\\
     &(Lat(\widetilde{\mathcal{N}})-\hat{L})*z \leq \sum\_{i=1}^m\sum\_{j=1}^nb^L\_{i,j}*b^{\widetilde{B}}\_{i,j},  (Eng(\widetilde{\mathcal{N}})-\hat{E})*z \leq \sum\_{i=1}^m\sum\_{j=1}^nb^E\_{i,j}*b^{\widetilde{B}}\_{i,j}  \\\\
    &\forall{1\leq i \leq m}, \sum\_{j=1}^nb^{\widetilde{B}}\_{i,j}=z, b^{\widetilde{B}}\_{i,j}\in\left\\{0,z\right\\}.
\end{split}
$$

### ILP Solving
To solve the ILP equations, we use the off-the-shelf Linprog Python package and the Gurobipy Python package to find feasible candidate solutions.
- Linprog is a basic integer programming solver that can be used on almost all edge devices, even on the resource-constrained Raspberry Pi. We use it to implement the branch and bound method and solve the ILP problem.
- Gurobipy is a more powerful solver, which has built-in a variety of advanced solving algorithms such as heuristic algorithms, and can flexibly utilize all available hardware resources on the device. Although Gurobipy is powerful, it requires more hardware resources than Linprog.

Therefore, for devices with limited hardware resources, we use Linprog for searching. For well-resourced devices, we use Gurobipy.

[9] Siegfried Schaible et al. Fractional programming. 1983.

---

### Decision · Program_Chairs · 2023-09-21

**Decision:**

Accept (poster)

**Comment:**

This study proposes a mathematical programming based NAS method named MathNAS, which is composed of three steps, block performance estimation, network performance prediction, and solving an integer linear programming problem to perform NAS based on the established mapping between block performance and network performance. After rebuttal, most reviewers acknowledged that the authors have addressed the concerns, and all reviewers suggested acceptance for this paper. The AC inspected the paper and all the discussions, and agreed that the paper should be accepted for the clarity of presentation, the novelty of the method, and the strong experimental performance.